# A holistic view of the dynamics of long-lived valley polarized dark excitonic states in monolayer WS$_2$

Xing Zhu[1,5], David R. Bacon[1,4,5], Vivek Pareek[1,5], Julien Madéo[1,5], Takashi Taniguchi[2], Kenji Watanabe[3], Michael K. L. Man[1] & Keshav M. Dani[1] ✉

With their long lifetime and protection against decoherence, dark excitons in monolayer semiconductors offer a promising route for quantum technologies. Optical techniques have previously observed dark excitons with a long-lived valley polarization. However, several aspects remain unknown, such as the populations and time evolution of the different valley-polarized dark excitons and the role of excitation conditions. Here, using time- and angle-resolved photoemission spectroscopy, we obtain a holistic view of the dynamics after valley-selective photoexcitation. By varying experimental conditions, we reconcile between the rapid valley depolarization previously reported in TR-ARPES, and the observation of long-lived valley polarized dark excitons in optical studies. For the latter, we find that momentum-dark excitons largely dominate at early times sustaining a 40% degree of valley polarization, while valley-polarized spin-dark states dominate at longer times. Our measurements provide the timescales and how the different dark excitons contribute to the previously observed long-lived valley polarization in optics.

In two dimensional (2D) semiconductors, the Coulomb interaction between the electron and hole leads to tightly bound excitons that exist even at room temperature. Moreover, in the case of transition metal dichalcogenides (TMDC)—prototypical 2D semiconductors, their honeycomb lattice structure creates two degenerate, but inequivalent valleys at the K and K' points at the edge of the Brillouin zone (BZ)[1]. For monolayer (1 L) TMDC, the lack of inversion symmetry enables valleytronics applications, with information encoded in the valley state of the bright excitons residing in the K- or K'-valley (Fig. 1a)[2,3]. Nonetheless, in these systems, the presence of additional nearly-degenerate spin- and momentum-dark excitons— those that do not interact with light due to the respective conservation rules, complicates the picture. Phonon interactions create momentum-dark excitons with the electron residing in the opposite K'(K) valley or Q valley from the exciton-bound hole[4,5], (Fig. 1a, b). Spin-dark excitons, that exist due to the presence of a relatively small spin-orbit split in the K (K') conduction band (Fig. 1b), can form by intravalley scattering mechanisms[6]. In addition to these interactions that scatter the bright excitons into optically inaccessible dark states, another primary impediment to valleytronics in 1 L TMDCs is the intervalley exchange interaction, which couples the K and K' valleys via a dipole-dipole interaction, flipping simultaneously electron and hole spins[7,8]. This results in the transfer of bright excitons from one valley into the other on a sub-100 fs timescale[9,10], rapidly depleting valley information initially encoded into the system[11–13]. Optically accessible interlayer excitons (ILX), found in heterobilayer systems, provide one possible way around the problem as they do not undergo intervalley exchange

[1]Femtosecond Spectroscopy Unit, Okinawa Institute of Science and Technology, Okinawa, Japan. [2]International center for Materials Nanoarchitectonics, National Institute for Materials Science, 1-1 Namiki, Tsukuba, Japan. [3]Research Center for Electronics and Optical Materials, National Institute for Materials Science, 1-1 Namiki, Tsukuba, Japan. [4]Present address: Department of Chemistry, University College London, London, United Kingdom. [5]These authors contributed equally: Xing Zhu, David R. Bacon, Vivek Pareek, Julien Madéo. ✉e-mail: KMDani@oist.jp

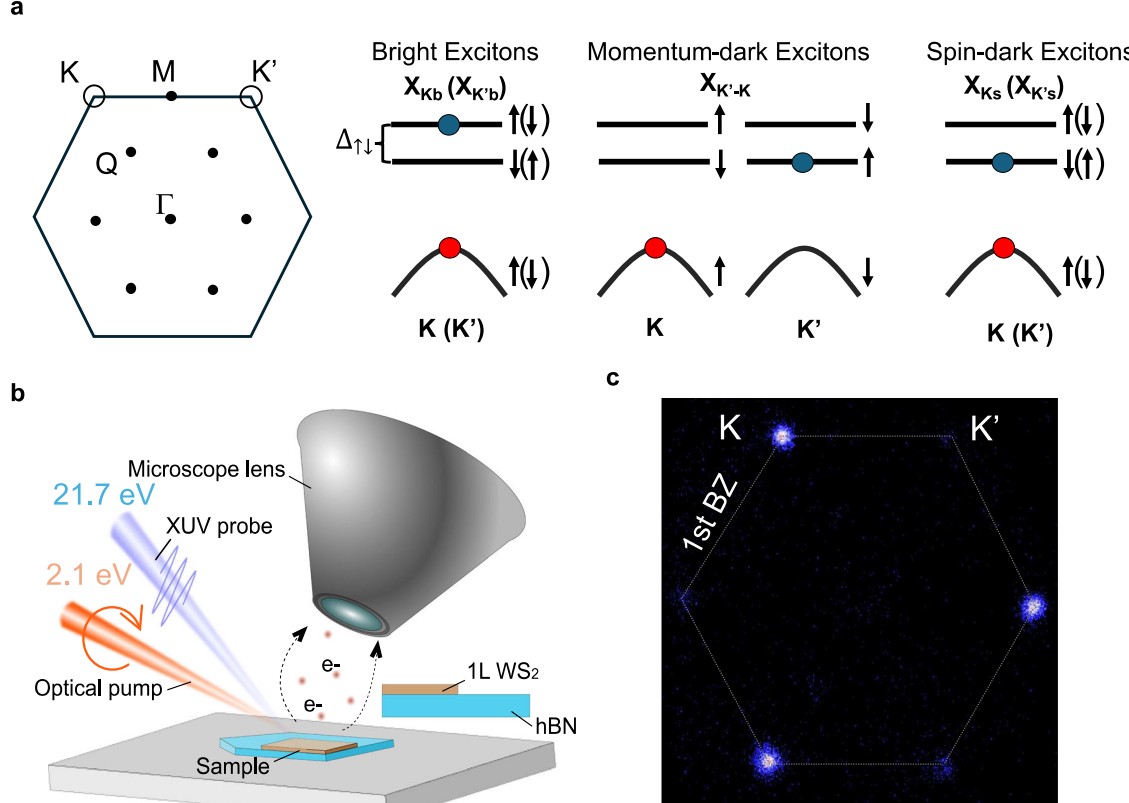

**Fig. 1 | Time-Resolved XUV ARPES of valley polarized excitons in monolayer WS$_2$. a** (left) Schematic depicting the hexagonal Brillouin zone (BZ) of monolayer WS$_2$ showing the K and K' valleys at the vertices of the BZ and intermediate $Q$ valley and (right) band diagram describing the bright excitons, the K'-K momentum-dark excitons and the spin-dark excitons. The red dots represent the hole in the valence band. The blue dots represent the location of the electron in spin-split states (dashed lines). The arrows represent the spin configuration in the K valley and arrows in brackets in the K' valley. **b** Simplified experimental setup using a circularly polarized photoexcitation and a XUV photoemission probe on a WS$_2$ monolayer sample on hBN to photoemit exciton-bound electrons that are collected by the lens of a momentum microscope. **c** ($k_x$,$k_y$) ARPES data (energy integrated between 1.9 to 2.2 eV above the valence band corresponding to the energy of the exciton-bound electron signal), at 0 ps time delay showing the valley contrast between the K and K' valley. A 120° rotating average centered at was performed to symmetrize the photoemission signal of each K and K' valleys (See SI §5).

interaction. However, the excitonic landscape in heterobilayer systems is even more complex, and includes the need for an intricate, non-resonant formation pathway for the ILX due to their weak oscillator strength[14]. Moreover, the need for precisely twisted additional layers creates challenges for future scalable device fabrication.

Another potential route to overcome the above challenges with bright excitons is to encode the valley information into a dark excitonic state in 1 L TMDC. It has been theoretically proposed that these would exhibit a strong elongation of the valley-polarization lifetime since they do not exhibit the intervalley exchange interaction between the two K and K' valleys[8]. Furthermore, due to their lack of interaction with electromagnetic radiation, dark excitons are expected to live longer[15–17] and decohere less[18,19], as compared to bright excitons. Previous optical experiments have confirmed the presence of long-lived valley-polarized dark excitons[20] in monolayer WSe$_2$, including momentum-dark excitons[21], spin-dark excitons[15] and dark trions[22]. Beyond this, several key aspects of valley-polarized dark excitons remain unknown—both in terms of their fundamental properties and their potential applications in quantum technologies. For instance, we lack information on the population of each of the different dark excitonic states at a particular time-delay relative to all the excitons that are generated after photoexcitation. Understanding which specie of valley-polarized dark excitons dominate at a given delay is important to being able to manipulate information stored in the valley degree of freedom of dark excitons. Furthermore, it is also unclear

whether experimental conditions can impact the relative contribution of a particular dark state to the overall photoexcited excitonic population and thereby affect the degree of valley polarization. Such information likely lies beyond the reach of conventional optical spectroscopy techniques.

Time- and angle-resolved photoemission spectroscopy (TR-ARPES)—a powerful technique to access the momentum character of excitons, their dynamics and the absolute excitonic populations[23–26] has the potential to answer these questions. However, prior TR-ARPES measurements on atomically thin TMDC did not observe the long-lived valley polarization seen with optical spectroscopy. Instead, they observed a rapid valley depolarization due to the intervalley exchange interaction[27], thus creating an apparent inconsistency between these two powerful experimental platforms.

In this letter, we perform time-resolved momentum microscopy on monolayer WS$_2$ with sufficient energy resolution to resolve the various spin- and momentum-dark excitonic states that form over the entire BZ after the photoexcitation of the valley-polarized bright excitons. Using a model based on rate equations, we extract the occupation of the various excitonic states and the timescales of intervalley exchange interaction, exciton-phonon scattering and intravalley spin relaxation. We find that under the experimental conditions of low temperature, low-intensity and resonant excitation, the intervalley exchange interaction of the initial K-valley-polarized bright exciton into the K'-valley is suppressed. Instead, one scatters almost exclusively into a specific, intermediate energy, valley-polarized

momentum-dark exciton. This dark exciton maintains its valley selectivity for several picoseconds, nearly two orders of magnitude longer than the bright exciton. In contrast, our measurements at room temperature or high excitation intensities show the more commonly expected behavior—the initial valley polarization vanishes within a few 100 s of fs, also seen in previous measurements for non-resonant excitation[27].

## Results

### Tr-ARPES experiment on 1 L WS₂: Valley-selective excitation

Our sample is an exfoliated WS₂ monolayer, transferred on a thin hBN buffer layer supported by a conducting Si substrate (see Methods). The TR-ARPES experiments (Fig. 1b) were conducted using a momentum microscope[28,29] as described in previous works[23,25,30]. Measurements were performed at 90 K, unless specified otherwise. With our current instrument capabilities and sample quality, we measure a FWHM linewidth of 88 meV of the top valence band in static ARPES (See SI §2). To resolve the valley dynamics of excitons, the sample was photoexcited with a 2.1 eV circularly polarized pump in resonance with the A-exciton to selectively populate the K-valley as shown in Fig. 1c. We photoexcited a low density of $4.5 \times 10^{11}$ cm$^{-2}$ excitons (see SI §6), thus limiting pump-induced band broadening effects. Thereby, in our experiments, the FWHM linewidth of the top valence band after photoexcitation and the photoexcited excitonic state did not exceed ~100 meV (See SI). As explained in more detail later, this enabled the resolution of the spin-bright and spin-dark excitonic states (Fig. 2a–e). We also carefully rotated our sample with respect to the XUV probe geometry to equalize the photoemission matrix elements between two adjacent K and K' valleys of the 1st BZ (See SI §4). This allows direct quantitative comparison of the photoexcited populations between the two valleys.

### Observation of valley-polarized momentum-dark excitons

First, let's discuss the observation of long-lived valley polarized momentum-dark excitons. To do so, we resolve in momentum space the constituent electrons and holes of excitons in both K and K' valleys (Fig. 2f). With the valley selective photoexcitation, at very early time-delays, we predominantly see (>90%) the bright K-valley excitons. This is evidenced by the large photoemission signal at the exciton energy from the exciton-electrons in the K valley and the corresponding presence of holes (loss of photoemission signal) in the valence band of the same valley (Fig. 2f–K valley). We also clearly observed the negative dispersion from the exciton-electron photoemission signal (Fig. 2a)–a hallmark of the excitonic state[30] (see SI §9). Additionally, at these early time-delays, we observe a weak signal in the K' valley at the energy of the bright exciton (see Fig. 2c). This is expected from the rapid intervalley exchange interaction of the photoexcited K-valley excitons into the K' valley (Fig. 2f–K' valley). We rule out any significant contribution from a momentum-dark state with electrons in the upper K' state as it requires a spin-flip scattering process (enhanced at higher temperature, see SI § 7). The weakly appearing $(k_x, k_y)$ momentum distribution of holes in Fig. 2f in the K' valley is due to too low experimental signal-to-noise for this low density ($\sim 7 \times 10^{10}$ cm$^{-2}$).

Strikingly, at 1 ps, we find that the dominant excitonic species is now a valley-polarized momentum-dark exciton (K'- K exciton), as seen by the large electron population in the K' valley and a large hole population remaining in the K valley (Fig. 2g). As expected from the momentum-dark K'-K exciton, the photoemission signal of the exciton-electrons is ~40 meV below the signal corresponding to bright excitons (see Fig. 2d). This value is similar to the intravalley spin-split observed in Fig. 2e and is also in good agreement with previous reports of the spin-splitting energy $\Delta_{\uparrow\downarrow}$[31,32]. We note that the energy of the momentum-dark exciton is expected to be slightly higher than the spin-dark exciton, due to electron-hole exchange repulsion, which reduces the binding energy of the spin-like momentum-dark exciton but does not impact the spin-unlike spin-dark exciton[33]. However, this

is not currently accessible with our energy resolution. At 1 ps, we also observe an exciton-electron signal in the K-valley, and the presence of holes in the K' valley (Fig. 2g). As discussed below, this is due to the presence of a weaker population of the opposite valley-polarized K-K' momentum-dark excitons, as well as the K-valley spin-dark excitons.

Our observations show the presence of a large population of valley polarized K'-K momentum-dark excitons up to long time-delays. This is surprising as one expected the intervalley exchange interaction to rapidly deplete valley polarization[12]. In our experiments, the low photoexcitation intensity plays a critical role in minimizing valley-depolarization due to the intervalley exchange interaction since it results in the creation of fewer excitons with non-zero center of mass momentum ($Q_{CM} \neq 0$) (This is also seen in the very clear negative dispersion of Fig. 2a that is exhibited by excitons with $Q_{CM} = 0$). Excitons with zero CM do not undergo intervalley exchange interaction[12,34], and hence, with the lower photoexcitation intensity, we get a smaller population scattering to the bright K' valley excitons. Besides the suppression of valley-depolarization, it is also surprising that valley polarization is preserved in a specific dark excitonic states, since one might have expected that the numerous excitonic scattering pathways would result in the formation of a large variety of excitonic species. The preservation of the valley polarization in a specific state makes it more feasible to control this polarization in future applications. We note that the long lifetime of the intermediate-energy K'-K excitonic state is not unexpected due to the potential bottleneck of spin-flip scattering suppressing the decay to the lowest energy dark excitonic state[35].

### Dynamics of the long-lived valley-polarized momentum-dark exciton

To effectively utilize the momentum-dark exciton in valleytronic applications, one must study its dynamics, as well as the global excitation dynamics, after valley-selective photoexcitation of bright excitons. To do so, we resolve the valley and spin states (via our energy resolution–Fig. 2e) of the exciton-bound electrons and holes over the entire BZ. The electron and hole populations are obtained by energy and momentum integrating their respective signals (see SI §3 and §6). From these measured electron and hole populations, we fit to a model based on rate equations, which enables us to extract the relevant excitonic populations and scattering timescales (see Fig. 3a and SI §7). The rate equations describing the temporal evolution of the various excitonic states have the general form[11]:

$$\frac{dX_n}{dt} = g_n(t) - \sum_m \frac{1}{\tau_{mn}} X_n + \sum_m \frac{1}{\tau_{nm}} X_m \tag{1}$$

Where $X_n$ is the density of the considered excitonic states, $g_n(t)$ is a generation term, $\tau_{mn}$ ($\tau_{nm}$) are the timescale corresponding to the depopulation of the $X_n$ ($X_m$) population by scattering to a $X_m$ ($X_n$) state including intervalley exchange ($\tau_{ex}$), exciton-phonon ($\tau_{ph}$), and intravalley ($\tau_{intra}$) scattering as well as recombination.

Our data reveals a clear sequential formation of the different excitonic states following the resonant excitation of the valley-polarized bright excitons (Fig. 3b) with an initial density of $4.5 \times 10^{11}$ cm$^{-2}$. First, only a small population of the bright K excitons ($7 \times 10^{10}$ cm$^{-2}$) rapidly scatters to the K' bright excitons through intervalley exchange interaction ($\tau_{ex} = 0.3$ ps), due to the low photoexcitation intensity, as discussed above. Following this, we see the predominant formation of the K'-K momentum-dark excitons (>60% at 1 ps) via intervalley phonon scattering ($\tau_{ph} = 0.8$ ps). Correspondingly, the initially photoexcited population of bright K excitons rapidly depletes (<0.5 ps). This K'–K momentum-dark exciton remains the dominant species across our experimental temporal range (10 ps) and maintains a high degree of valley polarization (>40%), defined as $P(X_{K'-K}) = \frac{nX_{K'-K} - nX_{K-K'}}{nX_{K'-K} + nX_{K-K'}}$, where nXi

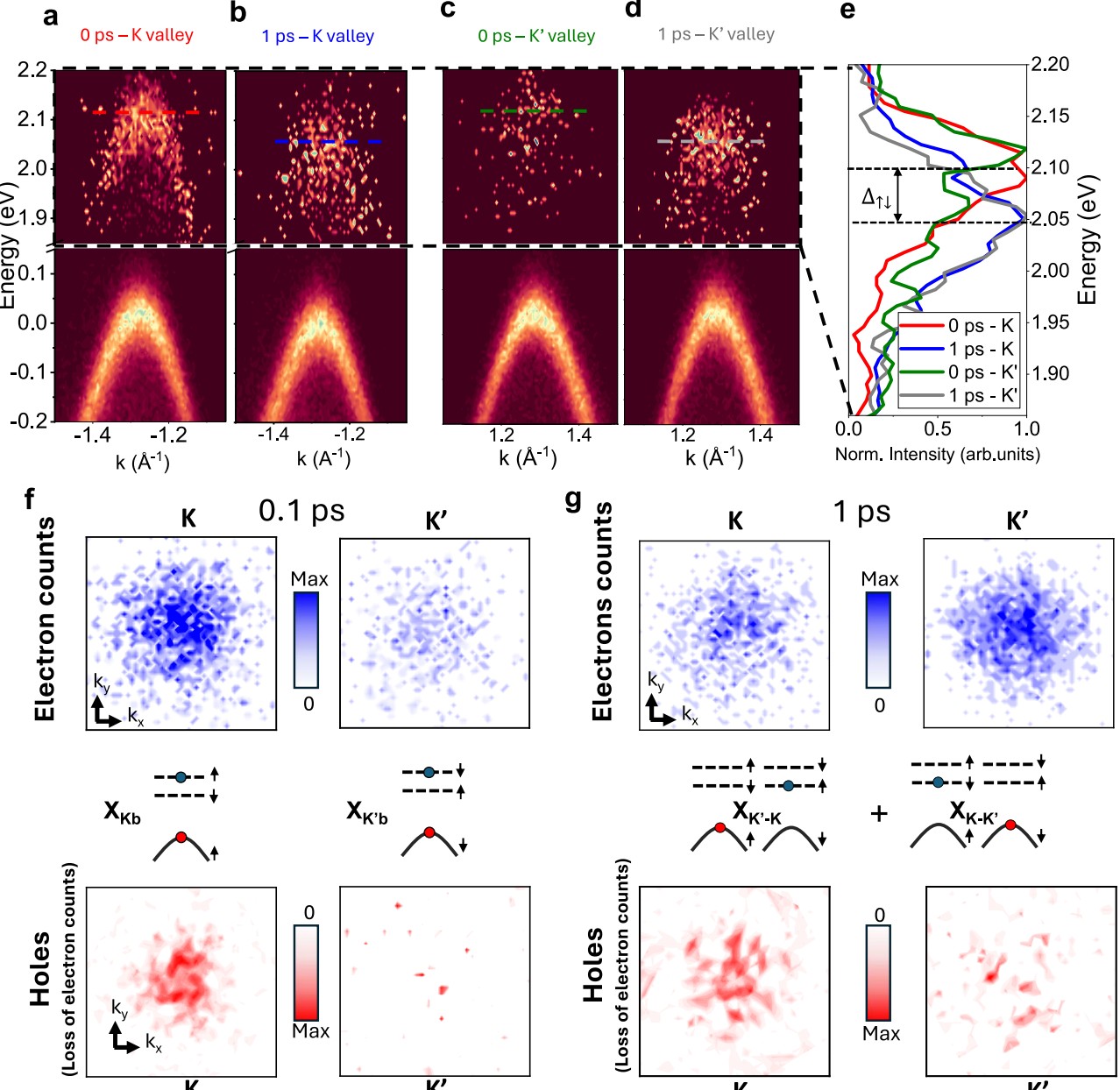

**Fig. 2 | Valley polarized momentum-dark excitons. a–d** Energy and momentum resolved linecut along the Γ-K-M axis showing. At 0 ps time delay, the resonantly photoexcited bright exciton signal in the K valley shows an exciton-bound electron with negative dispersion, located 2.1 eV above the valence band (red). By 1 ps, in the same valley, this electron signal has relaxed to a lower energy state (blue). In the K' valley, a weak electron population is observed at 0 ps at the photoexcitation energy of 2.1 eV (green). At 1 ps, it evolves into a much larger population that shows up at a lower energy (gray) (data around the exciton electron energy, 1.9–2.2 eV, were normalized at each k-vector). The corresponding energy distribution curves in (**e**) shows the energy difference between the bright exciton state which dominates at 0 ps and lower energy states that shows up at 1 ps. **f** Photoemission signals from

electrons around the A exciton energy and from holes at the valence band during photoexcitation (0.1 ps). For the electrons, the ARPES signals were energy integrated between 1.9 and 2.2 eV and displayed in $k_x,k_y$ momentum space. For the holes, we display the difference between negative time delay and after photoexcitation ARPES signals at the top of the valence band. The data were energy integrated over 100 meV (−0.05 to 0.05 eV) and a 120° rotating average around the center of the Γ valley was performed to clearly display the photoemission count loss corresponding to the presence of holes. **g** Photoemission signals from exciton bound electrons and holes around the A exciton energy at 1 ps using a similar analysis as in (**f**).

is the density of exciton Xi, through this time (Fig. 3c). In comparison, the degree of valley polarization of the bright exciton is less than 10% within a few hundred fs (Fig. 3c). We expect that this long-lived polarization is due to the lack of intervalley exchange interaction for the momentum-dark excitons[8], as well as the spin-flip or energy cost associated with the momentum-dark exciton scattering back into an intra-valley exciton.

Our simple model also allows us to extract the dynamics of the other excitonic states that form after valley-polarized photoexcitation and their associated scattering times (see Fig. 3b). In particular, the spin-dark excitons form with a much slower scattering time of a few ps, consistent with previous report of spin relaxation in W-based TMDCs[22,35]. Our data shows that the spin-dark excitons also carry a similar degree of valley selectivity, albeit with an order of magnitude

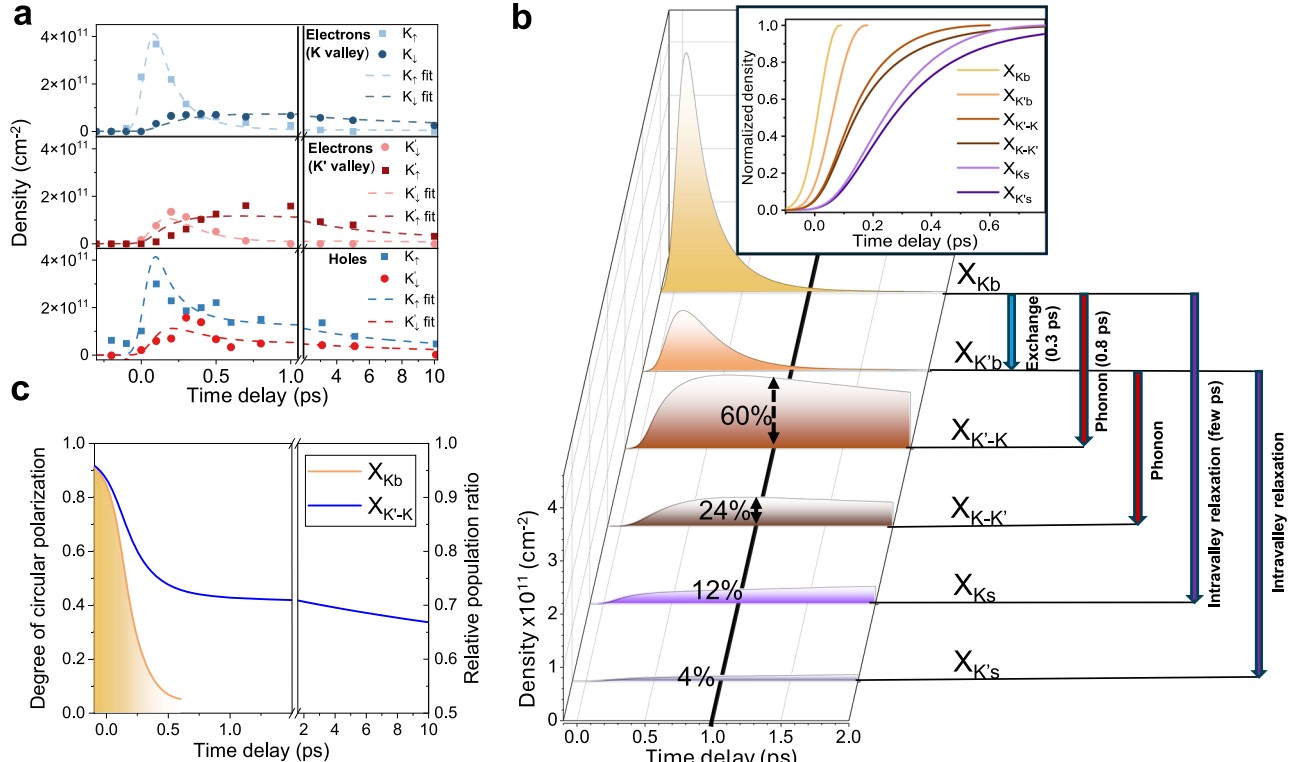

**Fig. 3 | Valley-polarized bright exciton to dark exciton scattering dynamics and distribution at low temperature and low exciton density. a** Measured dynamics of the electron density in the K and K' valley for each spin-split state and hole density in the valence band (dots). The dotted lines show the fit obtained from our model based on rate equations. **b** Bright ($X_{Kb}$ and $X_{K'b}$), momentum-dark ($X_{K'-K}$ and $X_{K-K'}$) and spin-dark ($X_{Ks}$ and $X_{K's}$) dominant exciton populations extracted from our model based on the experimental fit in (**a**). At 1 ps, we display each excitonic species

contribution to the total population. On the right, we present a diagram describing the formation and scattering processes associated to each excitonic populations. Inset: Normalized early time dynamics showing the formation sequence of each excitonic state. **c** Temporal evolution of the degree of circular polarization and relative population ratio for the $X_{Kb}$ bright excitons (yellow) and for the $X_{K'-K}$ and $X_{K-K'}$ momentum-dark excitons (blue).

lower density (see SI §8). We also note that although our energy resolution is insufficient to directly confirm that the spin-dark exciton is the lowest energy state, our observation that its population continues to increase over our measured time window (see SI §8) is consistent with this prediction.

We note that the above dynamics are under the specific photoexcitation conditions of low intensity ($4 \times 10^{11}$ cm$^{-2}$), low temperature (100 K) and resonant to the A-exciton. Increasing photoexcitation intensity (to a density of $2 \times 10^{12}$ cm$^{-2}$), sample temperature or photoexcitation energy leads to substantially different dynamics. In particular, we observe that the initially photoexcited degree of valley polarization is almost entirely depleted within a picosecond (Fig. 4), as also seen in previous experiments[27]. At higher temperature (and low intensity), by fitting our data with a model based on rate equation, we see that scattering processes that involve a spin flip are enhanced, leading to a rapid depolarization as seen in Fig. 4e, f and that additional scattering channel open that populate the Q-K momentum-dark excitons (see SI §7 for a detailed description of the dynamics and model based on rate equations). A higher intensity (Fig. 4c, d) provides excess center-of-mass momentum to the bright exciton that is expected to enhance of the intervalley exchange interaction[33] and lead to a faster depolarization. This is confirmed by measurements showing that increasing pump intensity leads to a reduced valley polarization of the bright exciton, accompanied by a broadening of the exciton–electron momentum distribution (See SI § 10).

## Discussion

Our results demonstrate that at low temperature, after a low intensity, resonant and valley-selective photoexcitation of bright excitons,

the valley-polarized population of momentum-dark excitons dominate (85% of the population at 1 ps) and with a 40% degree of valley polarization (for at least 10 ps). This provides important information towards achieving dark valleytronics, where the long-lived dark excitons that are naturally protected from decoherence and valley-depolarization, are used as an information carrier. Our work shows that, depending on experimental conditions, one can switch from a rapid depolarization process to the formation of long-lived valley-polarized dark excitons. Future research in methods to briefly and controllably brighten the momentum-forbidden dark excitons, e.g., with strain[36] or phonon-assisted mechanisms[4,5,21], as well as the influence of intervalley exchange interaction in these processes, would enable coherent initialization and read-out of the dark states – next steps in the development of dark excitons for quantum applications. In addition to the momentum-dark excitons, our work also indicates that spin-dark excitons also host a valley-polarized population for even longer times, although they represent only a small fraction of the initial population. Techniques utilizing magnetic field pulses[15] or surface plasmons[37] to briefly and controllably brighten the spin-dark excitons, and enhance their valley-polarized population, may offer an alternate viable path to using these excitonic species as well for future valleytronics applications. Finally, we note the important role that 1 L WS$_2$ may play in enabling the transfer of valley polarization from the bright to the momentum-dark excitons, due to the specific spin- and energy-ordering of the excitons in this system. Relatedly, a different substrate as well as doping may also influence the exciton dynamics and populations in each excitonic state. Future research in the dark exciton dynamics in other atomically thin semiconductors and their twisted heterostructures, with their

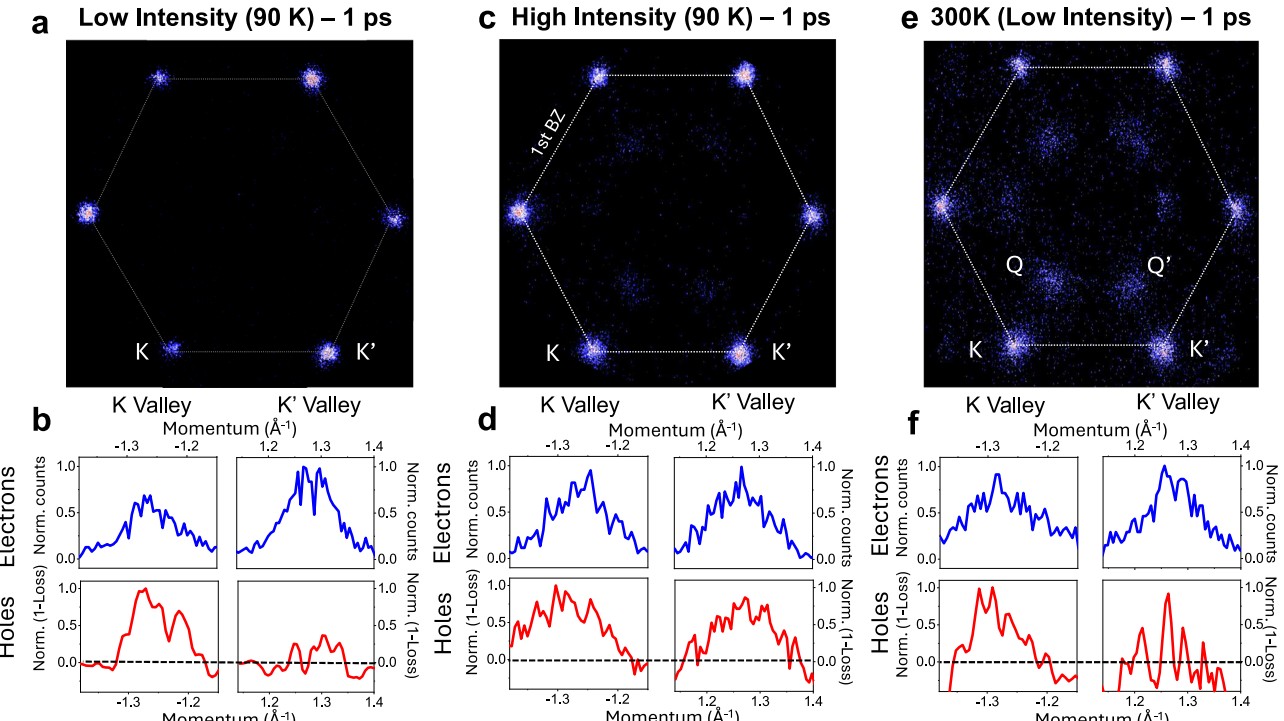

**Fig. 4 | Loss of valley polarization at high intensity and high temperature (300 K). a** Photoemission signal from the exciton-bound electrons at low intensity and low temperature at 1 ps. A 120° rotating average was performed to symmetrize the matrix element as in Fig. 1c. **b** Corresponding electron (blue) and hole (red) distributions. **c** Photoemission signal from the exciton bound electrons at high intensity at 1 ps **d** Corresponding electron (blue) and hole (red) distributions **e** Room temperature ARPES data at 1 ps and low intensity. **f** Room temperature and low intensity electron (blue) and hole (red) distributions.

unique spin- and energy-ordering, could lead to unexpected opportunities[38,39].

## Methods

### Sample fabrication

The studied sample is composed of a mechanically exfoliated monolayer $WS_2$ transferred onto a hBN thin layer. hBN is used to prevent exciton quenching from the n-doped Si substrate and to be consistent with its routine use in optical experiments. The hBN is directly transferred and cleaved on the Si substrate to obtain a pristine surface. The exfoliated $WS_2$ is transferred using the viscoelastic stamping method based on PDMS. After transfer, the sample is immediately rinsed in acetone and isopropanol followed by in-situ annealing at 350 °C for 11 h in the ultra-high vacuum preparation chamber of the momentum microscope.

### Time-resolved XUV-μ-ARPES

The experiment is driven by a high average power Yb:fiber amplifier (1030 nm, 250 fs, 100 μJ) operated at 1 MHz. 20 μJ are used to drive a noncollinear optical parametric amplifier tuned to the A exciton resonance (2.1 eV) of monolayer $WS_2$. A quarter waveplate is then used to create a circular polarization to photoexcite the sample. The XUV probe is based on gas-phase High-Harmonic Generation. A portion of the laser is frequency doubled using a BBO-crystal and 10 μJ are focused into a Kr gas jet to an intensity of $2 \times 10^{14} W/cm^2$. The resulting harmonic comb is then filtered by a set of Al and Sn foils to select the 21.7 eV harmonic with an estimated photon flux at sample of about $10^{11}$ ph/s. No measurable space charge effects were observed with the probe. Both pump and probe are focused on the sample located in the ultra-high vacuum chamber of a momentum microscope. A field aperture of 16 μm was positioned in the image plane of the microscope to transmit only photoelectrons from the

monolayer $WS_2$ area. The microscope is then set to project a magnified image of the back focal plane of the objective lens on a micro channel plate and the electron energy is measured using a time-of-flight detector.

## Data availability

The data that supports the finding of this work are available upon request to the corresponding author.

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

## Acknowledgements

This work was supported by the Femtosecond Spectroscopy Unit at the Okinawa Institute of Science and Technology Graduate University. J.M. acknowledges support from the JSPS KAKENHI (Grant Number 24K00561). K.M.D. acknowledges support from the JST FOREST (Grant number 23718777) and the JSPS KAKENHI (Grant numbers 22K18270, 24H00191 and 23K25807). K.W. and T.T. acknowledge support from the JSPS KAKENHI (Grant Numbers 21H05233 and 23H02052), the CREST (JPMJCR24A5), JST and World Premier International Research Center Initiative (WPI), MEXT, Japan. We thank the OIST engineering support section for their support. We thank M. Naik and O. Karni for insightful discussions.

## Author contributions

J.M., M.K.L.M. and K.M.D. designed the experimental setup. J.M., M.K.L.M., X.Z., V.P. and D.B. built the experimental setup. D.B., X.Z. and V.P. performed the experiments. J.M., D.B., X.Z. and V.P. analyzed the experimental data. D.B., M.K.L.M. and J.M. performed the rate equation analysis. V.P. and X.Z. prepared the sample. T.T. and K.W. provided high-quality hBN for sample preparation. K.M.D. supervised the project. All authors contributed to the manuscript.

## Competing interests

J.M., M.K.L.M. and K.M.D. are inventors on a granted patent related to this work filed by the Okinawa Institute of Science and Technology School Corporation (US patent 11,372,199). The authors declare no other competing interests.
