## [Transparent Peer Review file · Nature Communications]

A holistic view of the dynamics of long-lived valley polarized dark excitonic states in monolayer WS₂

Corresponding Author: Professor Keshav Dani

Version 0:

Reviewer comments:

Reviewer #1

(Remarks to the Author)

The manuscript entitled 'Creating long-lived valley-polarized dark excitons in a monolayer semiconductor' reported valley-polarized dark excitons, especially momentum-dark excitons of WS₂ monolayer by using time-dependent ARPES. This work is interesting regarding the method and topics, particularly time-resolved ARPES, which enables us to directly visualize the evolution of bright exciton in momentum space. However, I think the scientific impact of this manuscript may not reach the criterion of Nature Communications, and from my view, it's more suitable to a specialized journal. Here are some main comments:

(1) The scientific significance of this manuscript is not clear to me. The main result is that the momentum-dark excitons have high valley polarization together with a long valley lifetime. I think this result was already disclosed. I don't see significantly improved findings by using the advanced APERS tool. With respect to momentum-dark excitons, what new features can we find out compared to common optical methods? The authors have great previous works, such as Science 370, 1199-1204(2020), Nature 603, 247-252 (2022), what new advances in this manuscript are revealed compared to them. In this work, it is intriguing that bright excitons scatter more efficiently into momentum-dark excitons than spin-dark excitons, but I don't think it is enough for Nature Communications considering the significance and innovation in physics.

(2) Following the above comment, this work demonstrates valley-polarized dark excitons, but doesn't present ability to manipulate them. The reported content, is however already disclosed. And the creation method of valley-polarized dark excitons, which is realized by optical resonance excitation, is not unique to APERS. As far as I understand, we can inject them by usual ultrafast laser. So it is hard for me to find out the innovation of this manuscript.

There are also some minor issues remained to be solved.

(1) Please plot the insets of Figures 2, 3 bigger. Now it is hard to see them clearly.

(2) The excitation condition (low excitation density, low temperature and resonance excitation) is of importance for the observation of valley-polarized dark excitons. The authors should present the results in the same way for comparison. The plotting of low density, 90 K result is suggested to be added, with the same form as Fig. 4a, c.

(3) Dark excitons have long (valley-)lifetime and high degree of polarization due to the lack of exchange interaction. How can they be read out for 'dark valleytronics'? And once they are accessible in reading out, they have stronger exchange interaction, and it means they may lose their advantages in valley polarization.

(4) Why the probe energy is chosen as 21.7 eV?

Reviewer #2

(Remarks to the Author)

The manuscript by David R. Bacon et al. demonstrated that under specific experimental conditions bright excitons in two-dimensional semiconductors predominantly scatter into a specific valley-polarized dark excitonic state. This dark excitonic state retains its valley polarization for significantly longer durations compared to bright excitons, providing a potential pathway for developing dark valleytronics applications. Overall, this manuscript is well organized and written clearly. Please find the below comments that need to be addressed before considering for publication.

1. While the data presented in the manuscript is clear and well-structured, the novelty appears somewhat limited. The presence and dynamics of dark excitons have already been demonstrated through various methods (e.g., Science 370, 1199-1204 (2020); Nat Commun 13, 232 (2022); Nano Letters 2022, 22 (7), 3087-3094). The authors' findings could merit

publication in Nature Communications if the novelty of their results is further elucidated.

2. Do the authors have any estimation of the direct exciton population in the K' valley?
3. The role of phonon interactions, while acknowledged, is not deeply probed. A more detailed analysis of how specific phonon modes influence scattering pathways and excitonic lifetimes would provide additional insights.
4. The use of an hBN substrate introduces potential interfacial effects that may influence the excitonic dynamics. These effects are not discussed in detail, which could limit the interpretation of the results.

Reviewer #3

(Remarks to the Author)

In this manuscript, the authors present a time- and angle-resolved photoemission experiment on monolayer tungsten disulfide (WS₂). Using circularly polarized pump pulses, the authors excite a valley-polarized population of bright excitons and follow the scattering of these excitons into spin-dark and momentum-dark excitonic states during the relaxation process. By performing the experiment at low-temperature and low-fluence, the authors show that the valley-polarization is preserved in the momentum-forbidden excitonic state for much several picoseconds. This is significantly longer than studies at high-fluence and high temperature have shown.

First and foremost, the authors conduct a very nice experiment with high-quality data. The authors took care in balancing the photoemission matrix elements for the K and K' valleys, as well as for the 120 degree rotation symmetrisation for the show momentum maps. The energy resolution of 50 meV from the Fermi-edge is very impressive for an HHG-momentum-microscope setup. The unambiguous hole-like dispersion observed is also quite nice.

In terms of novelty, the manuscript is also special, particularly as previous measurements have been unable to realize a long-lived valley polarization. If the low excitation fluence is key for realizing valley polarization, it's likely that the 1MHz repetition rate is crucial for obtaining of the signal to noise necessary to perform the experiment at such low fluences. As I understand, there are few MM-HHG-TRARPES set ups operating at this repetition rate, which also makes this work unique among its peers.

For the above reasons, I can recommend the manuscript for publication, however, there are several ways the figures and the manuscript could be improved. In particular, parts of the analysis and interpretation require clarification. The conservation of spectral weight between the valence band and the excitonic states, I believe, is crucially important to the dynamics of the excitons fitted for Fig. 3, and should be demonstrated. (If spectral weight is not conserved, this should also be discussed and justified). I leave my detailed comments below for the authors' consideration.

General Comments:

While the flow of the text reads quite naturally, and data is nice, I find the presentation of the figures less nice. The inconsistent font and panel sizing makes some figures hard to see. Some panels (such as the data in Fig 2) are easy to see – just need to see intensity – but are unnecessarily huge, while the insets (which contain all the information) are tiny with tiny fonts and really difficult to see. And similarly with Fig. 3. The text also suffers from some inconsistent naming conventions (see below) and inconsistent use of hyphens.

Detailed Comments

Introduction

1. Line 66 (and others) The authors refer to “the exchange interaction” which rapidly depletes valley polarization. Later this is referred to as “intervalley exchange scattering”, and “exchange scattering” and “intervalley exchange interaction”, and “electron-hole exchange”. Such inconsistency causes some confusion. Given that this is a central point of the paper, I recommend the authors to give a brief description of this interaction in the introduction and stick to a consistent language throughout the text.
2. Similarly, “momentum-forbidden dark”, “momentum dark exciton”, “momentum-dark exciton” “dark exciton” “intervalley dark exciton” and “indirect exciton” is also inconsistently used. Given the wordiness often encountered, I recommend “spin-dark” and “momentum-dark” which gets the point (dark because of spin, and dark because of momentum) across in the fewest words.
3. Fig. 1 is very crowded and difficult to digest. The authors might consider splitting (and modifying) the conceptual figure to go with the introduction, about the excitons, and a separate figure talking about TR-ARPES signatures of excitons the experimental results.
 - a. Panel 1a is needlessly confusing, the authors aim to show bright, spin-forbidden dark, and momentum-forbidden intervalley excitons using shaded circles. Here are some questions I have.
 - i. What does the blue and red color mean? Is it meant to be spin polarization? If so why are the valence bands not blue and red, but a different kind of blue?
 - ii. Are the conduction bands spin polarized? Why are they green?
 - iii. Is the “K” in the middle of the image a label for the valence band on the left? Or a part of the caption for the excitons?
 - iv. The authors show the hole in the valence band with the little colored circles, but where are the electrons? Do the electrons belong in the different conduction bands shown and K and K'? If so, why are they not drawn there?
 - v. Why are there little red and blue dispersions below the valence band? What do these have to do with the excitons?
1. Of course, I understand that the authors are trying to link the different excitons to various photoemission signatures in TR-ARPES, but this is in no way clear in the figure caption or the main text. Hence, this diagram is incorrect and confusing. TR-ARPES experiment on 1L WS₂: Valley-selective excitation

1. Line 139: The authors write “We also clearly observed the negative dispersion from the exciton -electron photoemission signal (Fig. 1d) – a hallmark of the excitonic state.” While I agree the authors, it is known in photoemission that both excitonic spectral weight and laser-assisted photoemission (LAPE) can give rise to this spectral signature. While the excitonic state has finite relaxation timescales, the LAPE signal is only present during the pump excitation. Are the authors able to concretely exclude LAPE signal during pump excitation? Given the critical dependence of the fits on the TR-ARPES dynamics (particularly at time zero), this seems important to answer.
2. Line 140: The authors write “Additionally, at these early time-delays, we observe a weak signal in the K' valley at the energy of the bright exciton. This is expected from the rapid intervalley exchange scattering of the photoexcited K-valley excitons into the K' valley (Fig.2a-K' valley).”
 - a. It is not clear if these excitons in K' valley are bright or momentum-dark. Based on the inset of Fig. 2a, I believe the authors mean to communicate that the bright excitons in K scatter into bright excitons states in K', but if this is the case, why is there no depletion of holes in K'?
 - b. Based on the data that I see in Fig. 2a, it appears that there are bright excitons at K, and momentum dark excitons in K', with electrons at the K' and holes in K. This would explain why we do not see the depletion of holes at K.
 - i. Based on the dynamics in Fig. 3b, it seems that there should be a significant amount of K-K' indirect (I assume this is the momentum dark at K'? unclear) excitons even at 0.1 ps.
 - c. In Fig 2, the authors differentiate the intensity from bright and momentum-dark excitons by showing intensity at the excitonic energy and the depletion of intensity at the valence band. This analysis relies on the fact that a pump-induced change of the occupation function is the primary contributing factor to the photoemission intensity, and that the photoemission matrix element and spectral function have no explicit time dependence. Typically, the spectral weight is conserved, such that an integration of the population and depletion gives a zero (see studies of TR-ARPES on graphene and graphite, for instance). If one integrates the electron population and hole depletion at both the K and K' point, is the spectral weight conserved? If the excitons at K' are meant to be bright, then I would guess not? Why not? It would be nice to add consistent numbers to the colorbars here.
 - d. If the spectral weight is not conserved, is the ratio of depletion/population the same at K and K', as one might expect if they are both bright excitons?
3. Line 155: The authors write: “Strikingly, at 1 ps, we find that the dominant excitonic species is now a valley-polarized momentum forbidden exciton (K'- K exciton), as seen by the large electron population in the K' valley and a large hole population remaining in the K valley (Fig.2b).” While I do not disagree with this statement, the timescale of 0.1 ps and 1 ps is not so different. How do we know the ratio of bright excitons in K' with momentum-dark excitons at K'?
 - a. Since Fig 2 shows data that is integrated from 1.9 eV to 2.2 eV, Fig. 1d is crucial to the interpretation of the data shown in Fig. 2, such that they should really be put together in one figure. Also, the panel of (0.1 ps K' valley) is missing from the story, and would be important to the interpretation.
4. The colors in Fig 2 are also confusing:
 - a. Previously red and blue were used for spin polarization, now they are used for depletion and population? Or valence band and conduction band?
 - b. Why do the insets have different colors? Yellow, red and blue? Are they meant to connect to something?
 - c. The insets are hard to see.
 - d. Why is the integration region 300 meV for the excitonic states, but only 100 meV for the valence band?
5. Line 171: typo for “interaction”
6. Line 171: The authors attribute the lack of valley-depolarization to intervalley exchange, which does not affect zero-momentum excitons. Can the authors give more support to this argument, in particular from experiment? Are there any other possible explanations for this?

Dynamics of the long-lived valley polarized momentum dark exciton

1. The supplementary material mentions that the electron and hole populations are partitioned the K-up, K-down, K'-up and K'-down states. But does not mention which energy-momentum windows from the experimental data are integrated for these values, and this is quite crucial to the analysis. Could the authors provide some spectra and show which intensities are calculated from which regions?
2. Line 215: How is degree of circular polarization (DOCP) calculated? And is this related to valley polarization?
3. Figure 3 is also not entirely clear. In particular the insets are again small and the text is difficult to read. The diagram to the right of Fig. 3b is also not entirely clear with respect to the dual arrows. For example, is exchange + phonon interaction directly converting the bright exciton to the K' spin-dark exciton? Or is there an intermediate state in the K' bright exciton? Or is it both?
 - a. To alleviate some heavy unreadable text, the authors may consider giving a symbol to each exciton, for instance “Kb” for “bright exciton at K” and “K's” for “spin-dark exciton at K' ”.
4. Line 240: Given that photoexcitation intensity is crucial, do the authors have any fluence dependence that can show how the valley polarization evolves with the density of excited carriers? Such a dependence may give clues as to the interactions involved in valley-depolarization.

Version 1:

Reviewer comments:

Reviewer #1

(Remarks to the Author)

The authors have well responded to my comments. The novelty of this manuscript is now clearly demonstrated, and I would like to recommend its publication in Nature Communications.

Reviewer #2

(Remarks to the Author)

The authors have now provided a more explanation of the novelty of their work and clarified the specific knowledge gaps their study addresses, particularly in reconciling discrepancies between prior optical and TR-ARPES observations of dark exciton valley polarization. I appreciate these clarifications and the improvements made to the manuscript.

While the technical execution and analysis in this manuscript are strong enough, the scientific significance is not transformative on its own, relative to the expectations of Nature Communications. The central result — resolving time-dependent valley polarization dynamics of different dark exciton species via TR-ARPES — is certainly valuable, but remains incremental, building closely on prior work, including the authors' own.

For publication in a general-interest journal with a broad scientific scope, such as Nature Communications, a demonstration of dark exciton dynamics alone may not be sufficient. To meet the journal's criteria, I would expect the authors to go further and demonstrate some form of active control, tunability, or a compelling connection to practical device-level implementation using such long lived dark exciton, which would significantly enhance the manuscript's impact.

As it stands, the manuscript reads as a technically well-executed diagnostic study, which may be more appropriate for a specialized journal focused on ultrafast spectroscopy.

I encourage the authors to consider extending the scope of their work to demonstrate how these findings could be leveraged for real-world applications, in order to fully realize the potential impact and completeness of the study.

Reviewer #3

(Remarks to the Author)

The authors have replied satisfactorily to my concerns and I support the publication of this manuscript. Below I leave some comments for the authors to consider.

General Comments

- k_x and k_y should have subscripts.

Detailed Comments

- Fig 1c caption "Exciton-bound electron energy" sounds strange and doesn't really make sense. Perhaps the authors should quote the energy above the valence band and expand on what that energy region represents in the text.
- Line 142: "momentum dark" is missing a hyphen.
- This discussion of Fig. 2 is hard to follow because there are 9 panels in Fig. 2a and 8 panels in Fig. 2b. Could the authors provide some additional labels and/or color references to track which panels go together between a and b and refer to them in the text? For example (line 149): we observe a weak signal in the K' valley at the energy of the bright exciton (see Fig.2a, green dashed line)
- Fig 3 caption: XKs is not defined

REVIEWER RESPONSE

We thank the reviewers for their feedback. We are delighted that they appreciate the quality of our experimental work, and that our findings have the potential for publication in Nature Communications. We understand the central request of Reviewers 1 and 2 that we need to provide more clarity about the novelty of our work. Relatedly, we recognize that the current state-of-the art on valley-polarized dark excitons was insufficiently discussed.

Before providing a point-by-point reply to the Reviewer's comments, we believe it will be helpful to present a summary discussion about the novelty of our work, and the important advance it represents for the broader 2D semiconductor community. We then provide detailed point-by-point responses to the Reviewers.

GENERAL NOTE:

The reviewers' comments are shown in **bold**. For changes in the manuscript and SI, deleted text is shown in ~~red~~ and added text is shown in blue.

SUMMARY DISCUSSION ON THE NOVELTY OF THIS MANUSCRIPT

The Reviewers are correct that the observation of long-lived valley-polarized dark excitons has been disclosed before, particularly using optical spectroscopy. In monolayer (1L) WSe₂, momentum-dark states brightened by chiral phonons have shown a 40% degree of valley-polarization using photoluminescence experiments [ACS Nano 13, 14107-14113 (2019)]. In the same material, other studies have reported the observation of valley-polarization of spin-dark states using magneto-photoluminescence [Nature Nanotech. 12, 883-888 (2017)], with a 16% valley polarization measured at the phonon replica of the spin-dark state [Nature Comm. 10, 2469 (2019)]. Also in 1L WSe₂, 55% valley polarization was reported for dark trions in a gated sample with a waveguide using a similar optical experiment [Nature Comm. 10, 4047 (2019)]. A recent review summarizes well the various known aspects of valley polarized dark excitons [Nanophotonics 9, 1811-1829 (2020)].

Beyond these, several important aspects still remain unknown regarding the fundamental nature of valley polarized dark excitons, and the potential to use them for

technological applications. For example, though it is known that valley polarized momentum- and spin-dark excitons exist, their relative (and absolute) population contributions among all the excitonic excitations that are created after photoexcitation is unknown. How these relative contributions evolve over time, and which species of valley-polarized dark excitons dominate at what timescales is also unclear. Knowledge about the relative contributions, and the timescales at which different species of valley-polarized dark excitons dominate will be important to ultimately storing and manipulating information in the valley degree of freedom of dark states. Furthermore, whether and how experimental conditions impact the time-evolving populations of the different valley-polarized dark excitons is also unknown.

Obtaining the above information through optical spectroscopy is quite challenging. Instead, time- and angle-resolved photoemission spectroscopy (TR-ARPES) – a powerful technique to access the momentum character of excitons, their dynamics and their absolute populations [Science 370, 1199-1204 (2020); Nano Letters 21, 5867-5873 (2021); Nature 603, 247-252 (2022); Nature 608, 499-503 (2022)]- is well positioned to answer these questions. So far, most prior works using this technique have focused on the materials' response under a *linearly* polarized photoexcitation, including the work from our group mentioned by the Reviewers. To the best of our knowledge, only two valley-resolved TR-ARPES studies on TMDCs exist. The first [PRL 125, 216404 (2020)] is on bulk WSe₂, and hence not applicable to monolayer systems studied here, given significant band structure differences. The second [PRL 130 046202 (2023)] is a seminal valley-polarized TR-ARPES experiment on a monolayer TMDC. However, in contradiction to the optics experiments above, they reported the *absence of any long-lived valley polarized excitons* – bright or dark.

It is clear that not only significant knowledge gaps exist to further the understanding and technological applications of valley-polarized dark excitons, but there are fundamental inconsistencies between above-mentioned optics experiments and valley-polarized TR-ARPES experiments performed to date.

For the furtherment of dark valleytronics, and to reconcile the results from two powerful spectroscopic techniques, we realized that a TR-ARPES measurement needed to be done under the right experimental conditions, such as using valley polarized excitation, at low temperature and low intensity. Also, sufficient energy resolution would be needed to resolve spin-split exciton states, along with access to the momentum character of electrons and holes in all the valleys over the entire Brillouin zone. This would provide a holistic view of all excitons in momentum space, thereby providing the absolute (and relative) valley-polarized populations in all the different excitonic states, as a function of time-delay after photoexcitation.

In this work, we first developed the needed experimental capabilities – unprecedented energy resolution, extremely high signal to noise, very low excitation intensities, and low temperatures.

By performing TR-ARPES with these capabilities, under the right experimental conditions – low temperature, low photoexcitation intensity and resonant excitation -- we were able to observe long-lived valley polarized momentum- and spin-dark excitons, consistent with optics experiments. This put us in the position to fill the previously discussed knowledge gaps: We were able to get a holistic view of how photoexcited valley polarized bright excitons scatter into a variety of states over the whole Brillouin zone, and what fraction of those were valley-polarized momentum- and spin-dark excitons. At 1ps, the momentum dark excitons corresponded to 85% of the overall excitonic population with a 40% degree of valley polarization – the latter being consistent with optics [ACS Nano 13, 14107-14113 (2019)], and the former being new knowledge showing the conditions under which valley-polarized momentum-dark excitons can be made to dominate the excitation landscape. At long time delays (> 10 ps), our data indicates that the dominant excitonic specie switches to the spin-dark excitonic states with a comparable degree of valley polarization. The formation time of the valley polarized K'-K exciton is measured to be 0.8 ps and that of the valley polarized spin dark is 4 ps.

Additionally, by varying experimental conditions, we could switch between the observation of long-lived valley polarized dark excitons (as seen previously in optics) to the rapid valley depolarization (seen in previous TR-ARPES experiments). Our findings reveal how certain experimental conditions suppress the valley-depolarizing exchange interaction between bright excitons, thus allowing the creation of long-lived valley polarized dark excitons. In doing so, we reconcile the apparent inconsistency between the previous optics and TR-ARPES experiments. Reconciling the observations from these two powerful techniques brings significant value to future studies of valley polarized exciton dynamics in TMDs.

We acknowledge (and apologize) that we failed to convey this novelty in our work clearly before. We hope that with this explanation, and the corresponding changes in the manuscript outlined below, the novelty of our work is now clear.

Change in manuscript:

- To reflect the key message of our work, we have changed the title of our manuscript to “A holistic view of the dynamics of long-lived valley polarized dark excitonic states in monolayer WS₂”
- We have reworded the abstract to reflect the discussion above as follow:

With their long lifetime and protection against decoherence, dark excitons in monolayer semiconductors offer a promising route for quantum technology applications. Optical techniques have previously been used to show that dark excitons can sustain a long-lived valley polarization. However, several important aspects remain unknown regarding the fundamental nature of valley polarized dark excitons, such as the relative contributions of the different types of dark excitons to the degree of valley polarization, how these relative contributions evolve over time, and the role of excitation conditions therein. Here, using time- and angle-resolved photoemission spectroscopy with high energy resolution, we obtain a holistic view of the dynamics of valley-polarized bright excitons scattering into a variety of states over the entire Brillouin zone. By varying experimental conditions, we reconcile between reports from two different experimental platforms – we switch between the rapid valley depolarization reported previously in TR-ARPES experiments, and the ability to see long-lived valley polarized dark excitons observed in optical studies. For the latter, we find that the intervalley momentum-dark exciton dominates at early times, measuring 85% of the overall excitonic population at 1 ps time-delay, with a 40% degree of valley polarization, while our data indicates that valley polarized spin-dark states dominate the excited state population at much longer time delay (> 10 ps). Our measurements provide important information regarding the timescales and degrees to which different species of dark excitons dominate and contribute to the previously observed long-lived valley polarization in optics.

We have removed in the introduction the following:

~~Initial optical experiments show that some degree of valley information does get transferred to dark excitons after the photoexcitation of valley polarized bright excitons^{21,26}. However, one might reasonably expect this to be low, given experimental results that show a rapid initial loss of valley information via the exchange interaction²⁷. Furthermore, even if valley-polarized, one might expect that the bright exciton scatters into several types of valley-polarized excitonic states, making the subsequent manipulation of valley information largely unfeasible.~~

- We have added the following:

Previous optical experiments have confirmed the presence of long-lived valley-polarized dark excitons²⁰ in monolayer WSe₂, including momentum dark excitons²¹, spin-dark excitons¹⁵ and dark trions²². Beyond this, several key aspects of valley-polarized dark excitons remain unknown—both in terms of their fundamental properties and their potential applications in quantum technologies. For instance, we lack information on the population of each of the different dark excitonic states at a particular time-delay relative to all the excitons that are generated after photoexcitation. Understanding which species of valley polarized dark excitons dominate at a given delay is important to being able to manipulate

information stored in the valley degree of freedom of dark excitons. Furthermore, it is also unclear whether experimental conditions can impact the relative contribution of a particular dark state to the overall photoexcited excitonic population, and thereby affect the degree of valley polarization. Such information likely lies beyond the reach of conventional optical spectroscopy techniques.

Time- and angle-resolved photoemission spectroscopy (TR-ARPES) – a powerful technique to access the momentum character of excitons, their dynamics and the absolute excitonic populations²²⁻²⁵ – has the potential to answer these questions. However, prior TR-ARPES measurements on atomically thin TMDC did not observe the long-lived valley polarization seen with optical spectroscopy. Instead, they observed a rapid valley depolarization due to the intervalley exchange interaction²⁷, thus creating an apparent inconsistency between these two powerful experimental platforms.

- In the conclusion, we have rephrased the following:

Our results demonstrate that at low temperature, after a low intensity, resonant and valley-selective photoexcitation of bright excitons, the valley-polarized population of momentum-dark excitons dominate (85% of the population at 1 ps) and with a 40% ~~DOCP~~ degree of valley polarization (for at least 10ps). This provides important information towards achieving dark valleytronics, where the long-lived dark excitons that are naturally protected from decoherence and valley-depolarization, are used as an information carrier.

Below, we provide the point-by-point reply to the Reviewer's comments.

Reviewer #1

The manuscript entitled ‘Creating long-lived valley-polarized dark excitons in a monolayer semiconductor’ reported valley-polarized dark excitons, especially momentum-dark excitons of WS₂ monolayer by using time-dependent ARPES. This work is interesting regarding the method and topics, particularly time-resolved ARPES, which enables us to directly visualize the evolution of bright exciton in momentum space. However, I think the scientific impact of this manuscript may not reach the criterion of Nature Communications, and from my view, it’s more suitable to a specialized journal. Here are some main comments:

(1) The scientific significance of this manuscript is not clear to me. The main result is that the momentum-dark excitons have high valley polarization together with a long valley lifetime. I think this result was already disclosed. I don’t see significantly improved findings by using the advanced APERS tool. With respect to momentum-dark excitons, what new features can we find out compared to common optical methods? The authors have great previous works, such as Science 370, 1199-1204(2020), Nature 603, 247–252 (2022), what new advances in this manuscript are revealed compared to them. In this work, it is intriguing that bright excitons scatter more efficiently into momentum-dark excitons than spin-dark excitons, but I don’t think it is enough for Nature Communications considering the significance and innovation in physics.

We hope to have now clarified to the Reviewer the novelty of our work in the ‘Summary Discussion of the Novelty’. We believe the reconciliation of the previously observed contradictions between two powerful experimental platforms for 2D semiconductors, the understanding of the physics that leads to long-lived valley-polarized dark excitons (seen previously in optics) versus rapid depolarization observed in TR-ARPES, the measurement of which species of dark excitons dominate at different timescales and how they contribute to the degree of valley polarization are all important advances that will further the ability to utilize the valley degree of freedom of dark excitons. We are confident of the significance of our work to the broader 2D semiconductor community and to the Nature Communications readers.

(2) Following the above comment, this work demonstrates valley-polarized dark excitons, but doesn’t present ability to manipulate them. The reported content, is

however already disclosed. And the creation method of valley-polarized dark excitons, which is realized by optical resonance excitation, is not unique to APERS. As far as I understand, we can inject them by usual ultrafast laser. So it is hard for me to find out the innovation of this manuscript.

We agree that creation of valley-polarized dark excitons is not unique to TR-ARPES. In our work, we have extended the capabilities of TR-ARPES, to operate under similar conditions as optical spectroscopy, at low intensity and low temperature, and this enabled the observation of valley-polarized dark excitons that have been observed previously in optics experiments. However, in contrast to optics, TR-ARPES can provide a holistic view of the dynamics of the valley-polarized states, the population of each state as a function of time delay, and the timescale of the scattering rates involved.

By using this technique, we find that K-K' momentum-dark excitons are the dominant species at early time delay while our data indicate that, at long time delay, spin-dark state dominate. All this information cannot be obtained by optical spectroscopy and is important to advance towards practical applications using dark excitons for technological applications.

There are also some minor issues remained to be solved.

(1) Please plot the insets of Figures 2, 3 bigger. Now it is hard to see them clearly.

As part of the reply to Reviewer 3, we have now reformatted all the figures of the manuscript for clarity.

(2) The excitation condition (low excitation density, low temperature and resonance excitation) is of importance for the observation of valley-polarized dark excitons. The authors should present the results in the same way for comparison. The plotting of low density, 90 K result is suggested to be added, with the same form as Fig. 4a, c.

We have now added in Fig.4 the low intensity and low temperature data showing, at 1 ps, the conservation of valley polarization in the momentum-dark exciton. In contrast to the high intensity (Fig.4c,d) and high temperature (Fig.4e,f), showing that at 1 ps, both electrons and holes population are balanced between the K and K' valleys, the low temperature and low intensity data (Fig.4a,b) shows the imbalance of population of electrons and holes between the K and K' valleys with a the majority of electrons in the K' valley and most of the holes remaining in the K valley.

Fig.R1: Revised Fig.4: Loss of Valley polarization at high intensity and high temperature (300K). (a) Photoemission signal from the exciton bound electrons at low intensity and low temperature at 1 ps. A 120° rotating average was performed to symmetrize the matrix element as in Fig.1c. (b) Corresponding electron (blue) and hole (red) distributions (c) Photoemission signal from the exciton bound electrons at high intensity at 1 ps. (d) Corresponding electron (blue) and hole (red) distributions (e) Room temperature ARPES data at 1 ps and low intensity. (f) Room temperature and low intensity electron (blue) and hole (red) distributions.

Change in manuscript:

- Panels 4e and 4f have been added to Fig.4
- Figure caption has been updated

(3) Dark excitons have long (valley-)lifetime and high degree of polarization due to the lack of exchange interaction. How can they be read out for ‘dark valleytronics’? And once they are accessible in reading out, they have stronger exchange interaction, and it means they may lose their advantages in valley polarization.

For dark valleytronics, one needs a coherent and ultrafast way to brighten dark excitons so they can be read out optically. Various approaches can be implemented, e.g. using phonon-

assisted mechanisms such as chiral phonons [ACS Nano 13, 14107-14113 (2019)] for momentum dark excitons or using photonics bound states in the continuum for spin-dark excitons [Nat. Comm. 13, 6916 (2022)].

Regarding the effect of intervalley exchange, our work shows that sufficiently low intensity suppresses such interaction for bright excitons (see also Fig. R8 and associated reply to Reviewer#3) as intervalley exchange interaction is only active for excitons with finite center-of-mass (COM) momentum [PRL 115, 176801; Phys. Rev. Rev. 2, 023322 (2020)]. For example, a low exciton density of bright excitons that form after scattering from a dark exciton is also expected to have minimum COM momentum and therefore to exhibit minimum depolarization. Nonetheless, future investigations should investigate the role of intervalley exchange interaction in the various methods used to brighten dark excitons such as phonon-assisted recombination of indirect excitons or strain.

Change in manuscript:

- In the conclusion, we have added the following:

Future research in methods to briefly and controllably brighten the momentum-forbidden dark excitons, e.g. with strain ³⁷ or phonon-assisted mechanisms ^{5,21,38}, as well as the influence of intervalley exchange interaction in these processes, would enable coherent initialization and read-out of the dark states – next steps in the development of dark excitons for quantum applications.

(4) Why the probe energy is chosen as 21.7 eV?

The probe energy used is the same as in our previous work and is related to experimental and technical optimization. We use the HHG nonlinear process to generate a XUV harmonic comb using a 515 nm driver with long pulses (250 fs). In this configuration, the highest harmonic photon flux is obtained at 21.7 and 27 eV. For TR-ARPES, we need to spectrally select a single harmonic. This can be done using a monochromator (more complex in alignment and lossy) or a simple and well-established technique [Nat. Comm. 6, 7459 (2015)] using a metal foil Sn bandpass filter at 22 eV. The latter allows for higher XUV throughput of the beamline as well as better long-term stability as fewer mechanical parts are used.

Reviewer #2:

The manuscript by David R. Bacon et al. demonstrated that under specific experimental conditions bright excitons in two-dimensional semiconductors predominantly scatter into a specific valley-polarized dark excitonic state. This dark excitonic state retains its valley polarization for significantly longer durations compared to bright excitons, providing a potential pathway for developing dark valleytronics applications. Overall, this manuscript is well organized and written clearly. Please find the below comments that need to be addressed before considering for publication.

1. While the data presented in the manuscript is clear and well-structured, the novelty appears somewhat limited. The presence and dynamics of dark excitons have already been demonstrated through various methods (e.g., *Science* 370, 1199–1204 (2020); *Nat Commun* 13, 232 (2022); *Nano Letters* 2022, 22 (7), 3087–3094). The authors' findings could merit publication in *Nature Communications* if the novelty of their results is further elucidated.

We are delighted that the Reviewer thinks our work could merit publication. In response to the Reviewer's concern regarding novelty, we have revised the manuscript and in the 'Summary Discussion on the Novelty of the Manuscript' at the beginning, more clearly articulate the unique aspects of our findings. Specifically, relative to the works mentioned above, in this manuscript we discuss the dynamics of *valley-polarized* dark excitons. While long-lived valley polarized dark excitons were reported using optical spectroscopy, previous time-resolved ARPES experiments reported the absence of any long-lived valley polarization. In this manuscript, by changing the experimental conditions we reconcile these two contradictory observations from two important experimental platforms and explain the associated physics. With respect to the optics experiments, we obtain a holistic view of the dynamics over the entire BZ and thereby report on the dynamics of the different types of dark excitons, their contribution to the degree of valley polarization, and how experimental conditions can impact these contributions.

2. Do the authors have any estimation of the direct exciton population in the K' valley?

As shown in Fig.3 of the manuscript, we have performed a fitting of our experimental data of the dynamics of the electron population in the spin-split states and the hole population in the valence band, allowing to extract the temporal evolution of the absolute population in each excitonic states. From this, we see that the maximum population of the K' bright exciton

reaches $7 \times 10^{10} \text{ cm}^{-2}$, so about 7 times less than the resonantly photoexcited K bright exciton. In reply to Reviewer#3 point 3a, we have added in Fig.2, the early time ARPES data showing the K' valley population.

Change in manuscript:

- The beginning of the last paragraph of page 9 has been changed:

Our data reveals a clear sequential formation of the different excitonic states following the resonant excitation of the valley-polarized bright excitons (Fig.3b) with an initial density of $4.5 \times 10^{11} \text{ cm}^{-2}$. First, only a small population of the bright K excitons ($7 \times 10^{10} \text{ cm}^{-2}$) rapidly scatters to the K' bright excitons through intervalley exchange interaction ($\tau_{ex} = 0.3 \text{ ps}$), due to the low photoexcitation intensity, as discussed above.

3. The role of phonon interactions, while acknowledged, is not deeply probed. A more detailed analysis of how specific phonon modes influence scattering pathways and excitonic lifetimes would provide additional insights.

We appreciate the reviewer's comment to provide an analysis of the phonon modes involved in the scattering pathways. In our work, we provide the timescale of the scattering to each excitonic states, but the specific phonon modes related to the scattering to the dark excitons is not directly accessible by our experiment, is beyond the scope of our current study and has been already extensively discussed elsewhere, for example [Nat. Comm. 10, 2469 (2019); PRB 106, 085414 (2022)].

Nonetheless, we provide new additional data and analysis performed at room temperature (under the same excitation conditions) in order to probe how the higher temperature affects the dynamics, involving enhanced phonon scattering, and in particular leads to a rapid loss of valley polarization (see Fig.R2 below).

In contrast to what we observed at low temperature, we find that scattering with spin flip now occurs both for holes and electrons. In the figure below, we show the experimental data at room temperature and the fit using rate equations. Within 1ps, we see that the electron populations in $K\downarrow$ and $K'\uparrow$ equalize, and we make the same observation for the holes of the K and K' valleys. Using our model, we extract the rates of each of those additional scattering processes and have added a complete description in the SI §7. In contrast to the low temperature data in which no Q valley momentum-dark excitons were measured, here we find a substantial population of Q valley excitons that are expected to form when phonon scattering is present, which we have previously reported in [Science 370, 1199-1204 (2020)].

Fig.R2: (a) Experimental dynamics of the spin-split electrons and holes at 300 K and low intensity. (b) Exciton dynamics at room temperature.

Change in manuscript:

- We have added a discussion above Fig.4:

At higher temperature (and low intensity), by fitting our data with a model based on rate equation, we see that scattering processes that involve a spin flip are enhanced, leading to a rapid depolarization as seen in Fig 4e and 4f and that additional scattering channel open that populate the Q-K momentum dark excitons (see SI §7 for a detailed description of the dynamics and model based on rate equations).

Change in SI:

- Model based on rate equation at room temperature has been added in section 7
- Figures showing the exciton dynamics at room temperature have been added in section 7

4. The use of an hBN substrate introduces potential interfacial effects that may influence the excitonic dynamics. These effects are not discussed in detail, which could limit the interpretation of the results.

Here, we used hBN for two main reasons: (1) the use of hBN is pivotal to perform high quality TR-ARPES and achieve high energy resolution due to its high flatness and because its insulating nature prevents exciton quenching from the conductive doped Si (required to avoid charging in the electron microscope) and (2) to be in par with the community as hBN is routinely used in the recent literature for device fabrication and for fundamental studies of 2D semiconductors as it provides better linewidths and quantum efficiency in optical spectroscopy compared to other substrates or even suspended monolayers [Sci. Rep. 6, 20890 (2016)].

Nonetheless, the Reviewer is correct that the use of different substrates or doping, for example, could affect the exciton dynamics. Future studies will determine the impact of substrate and doping on the overall evolution and populations of valley polarized excitons.

Change in manuscript:

- In the conclusion, we have added:

Finally, we note the important role that 1L WS₂ may play in enabling the transfer of valley polarization from the bright to the momentum-dark excitons, due to the specific spin- and energy-ordering of the excitons in this system. Relatedly, a different substrate as well as doping may also influence the exciton dynamics and populations in each excitonic state.

- In the methods section “sample fabrication”, we have added:

The studied sample is composed of a mechanically exfoliated monolayer WS₂ transferred onto a hBN thin layer. hBN is used to prevent exciton quenching from the n-doped Si substrate and to be consistent with its routine use in optical experiments.

Reviewer #3:

In this manuscript, the authors present a time-and angle-resolved photoemission experiment on monolayer tungsten disulfide (WS₂). Using circularly polarized pump pulses, the authors excite a valley-polarized population of bright excitons and follow the scattering of these excitons into spin-dark and momentum-dark excitonic states during the relaxation process. By performing the experiment at low-temperature and low-fluence, the authors show that the valley-polarization is preserved in the momentum-forbidden excitonic state for much several picoseconds. This is significantly longer than studies at high-fluence and high temperature have shown.

First and foremost, the authors conduct a very nice experiment with high-quality data. The authors took care in balancing the photoemission matrix elements for the K and K' valleys, as well as for the 120 degree rotation symmetrisation for the show momentum maps. The energy resolution of 50 meV from the Fermi-edge is very impressive for an HHG-momentum-microscope setup. The unambiguous hole-like dispersion observed is also quite nice.

In terms of novelty, the manuscript is also special, particularly as previous measurements have been unable to realize a long-lived valley polarization. If the low excitation fluence is key for realizing valley polarization, it's likely that the 1MHz repetition rate is crucial for obtaining of the signal to noise necessary to perform the experiment at such low fluences. As I understand, there are few MM-HHG-TRARPES set ups operating at this repetition rate, which also makes this work unique among its peers.

For the above reasons, I can recommend the manuscript for publication, however, there are several ways the figures and the manuscript could be improved. In particular, parts of the analysis and interpretation require clarification. The conservation of spectral weight between the valence band and the excitonic states, I believe, is crucially important to the dynamics of the excitons fitted for Fig. 3, and should be demonstrated. (If spectral weight is not conserved, this should also be discussed and justified). I leave my detailed comments below for the authors' consideration.

General Comments:

While the flow of the text reads quite naturally, and data is nice, I find the presentation of the figures less nice. The inconsistent font and panel sizing makes some figures hard

to see. Some panels (such as the data in Fig 2) are easy to see – just need to see intensity – but are unnecessarily huge, while the insets (which contain all the information) are tiny with tiny fonts and really difficult to see. And similarly with Fig. 3. The text also suffers from some inconsistent naming conventions (see below) and inconsistent use of hyphens.

Detailed

Comments

Introduction

1. Line 66 (and others) The authors refer to “the exchange interaction” which rapidly depletes valley polarization. Later this is referred to as “intervalley exchange scattering”, and “exchange scattering” and “intervalley exchange interaction”, and “electron-hole exchange”. Such inconsistency causes some confusion. Given that this is a central point of the paper, I recommend the authors to give a brief description of this interaction in the introduction and stick to a consistent language throughout the text.

We thank the reviewer for pointing the language inconsistencies in our manuscript. We have corrected them and we have added a brief description of the intervalley exchange interaction in the introduction.

Change in manuscript:

- In the 1st paragraph of the introduction, we have added:

In addition to these interactions that scatter the bright excitons into optically inaccessible dark states, another primary impediment to valleytronics in 1L TMDCs is the **intervalley exchange interaction which couples the K and K' valleys via a dipole-dipole interaction, flipping simultaneously electron and hole spins**^{7,8}. **Scattering** This results in the transfer of bright excitons from one valley into the other on a sub-100 fs timescale^{9,10}, **the exchange interaction** rapidly **depletes** **depleting** valley information initially encoded into the system^{11,12,13}.

2. Similarly, “momentum-forbidden dark”, “momentum dark exciton”, “momentum-dark exciton” “dark exciton” “intervalley dark exciton” and “indirect exciton” is also inconsistently used. Given the wordiness often encountered, I recommend “spin-dark” and “momentum-dark” which gets the point (dark because of spin, and dark because of momentum) across in the fewest words.

We agree with the reviewer that those inconsistencies led to confusion. We are now using the terms spin-dark and momentum-dark.

3. Fig. 1 is very crowded and difficult to digest. The authors might consider splitting (and modifying) the conceptual figure to go with the introduction, about the excitons, and a separate figure talking about TR-ARPES signatures of excitons the experimental results.

a. Panel 1a is needlessly confusing, the authors aim to show bright, spin-forbidden dark, and momentum-forbidden intervalley excitons using shaded circles. Here are some questions I have.

i. What does the blue and red color mean? Is it meant to be spin polarization? If so why are the valence bands not blue and red, but a different kind of blue?

ii. Are the conduction bands spin polarized? Why are they green?

iii. Is the “K” in the middle of the image a label for the valence band on the left? Or a part of the caption for the excitons?

iv. The authors show the hole in the valence band with the little colored circles, but where are the electrons? Do the electrons belong in the different conduction bands shown and K and K'? If so, why are they not drawn there?

v. Why are there little red and blue dispersions below the valence band? What do these have to do with the excitons? Of course, I understand that the authors are trying to link the different excitons to various photoemission signatures in TR-ARPES, but this is in no way clear in the figure caption or the main text. Hence, this diagram is incorrect and confusing.

Reply to 3.i - 3.v

We thank the reviewer for their constructive suggestion to improve the clarity of our figures. We have replaced and simplified the panel of Fig.1A according to the Reviewer's suggestions.

Fig.R3: Revised Fig. 1: Time-Resolved XUV ARPES of valley polarized excitons in monolayer WS_2 . (left) Schematic depicting the hexagonal Brillouin zone (BZ) of monolayer WS_2 showing the K and K' valleys at the vertices of the BZ and intermediate Q valley and (right) band diagram describing the bright excitons, the $K'-K$ momentum dark exciton and the spin-dark excitons. The red dots represent the hole in the valence band. The blue dots represent the location of the electron in spin-split states (dashed lines). The arrows represent the spin configuration in the K valley and arrows in brackets in the K' valley. (b) Simplified experimental setup using a circularly polarized photoexcitation and a XUV photoemission probe on a WS_2 monolayer sample on hBN to photoemit exciton-bound electrons that are collected by the lens of a momentum microscope. (c) (k_x, k_y) ARPES data, energy integrated at the exciton-bound electron energy (1.9 to 2.2 eV), at 0 ps time delay showing the valley contrast between the K and K' valley. A 120° rotating average centered at Γ was performed to symmetrize the photoemission signal of each K and K' valleys (See SI §5).

Change in manuscript:

- Figure 1 has been replaced
- Figure 1 caption has been updated

TR-ARPES experiment on 1L WS2: Valley-selective excitation

1. Line 139: The authors write “We also clearly observed the negative dispersion from the exciton -electron photoemission signal (Fig. 1d) – a hallmark of the excitonic state.” While I agree the authors, it is known in photoemission that both excitonic spectral weight and laser-assisted photoemission (LAPE) can give rise to this spectral signature. While the excitonic state has finite relaxation timescales, the LAPE signal is only present during the pump excitation. Are the authors able to concretely exclude LAPE signal during pump excitation? Given the critical dependence of the fits on the TR-ARPES dynamics (particularly at time zero), this seems important to answer.

The Reviewer is correct that one should pay attention to LAPE, i.e. replicas of the band structure due to the interaction of the photoemitted electrons with the pump field during photoexcitation. For the relevant data presented in our manuscript (as in revised Fig.2, see reply to point 3 below), no LAPE was observed in the low intensity regime ($4.5 \times 10^{11} \text{ cm}^{-2}$) (Fig. R4a). In order to observe band replicas originating from LAPE, we need to reach a very high pump intensity corresponding to a density of $\sim 6 \times 10^{12} \text{ cm}^{-2}$ with a resonant and linearly polarized excitation (see Fig.R4b) where only a weak hint of a replica at Γ is seen. We could observe a significant contribution from LAPE with a below resonance pump (1.9 eV), linear p-polarization and a high intensity (Fig.R4c). We can therefore confidently rule out any contribution of LAPE in the low intensity data presented in the manuscript.

Additionally, to convince the Reviewer that the negative dispersion does not only appear at zero-time delay and is not an effect of LAPE in the low intensity regime, we show below in Fig.R5 the negative dispersion both in the K and K' valley at a time delay of 5 ps, long after the photoexcitation. We have added the figures with the negative dispersion in the SI section 9.

Fig.R4: Experimental band structure at 0 ps time-delay (a) for Low intensity and resonant circularly polarized photoexcitation, (b) for high intensity and resonant linearly polarized photoexcitation and (c) for high intensity, below resonance and p-polarized photoexcitation.

Fig.R5: Data at 5 ps time-delay showing the valence band and the exciton electron signals (normalized at each k vector). The dispersion was obtained by fitting with a Gaussian function the energy distribution at each k vector and plotting the peak of each Gaussian.

Change in SI:

- Fig.S9 has been added to a new section (9) of the SI discussing the negative dispersion at long time delay

2. Line 140: The authors write “Additionally, at these early time-delays, we observe a weak signal in the K’ valley at the energy of the bright exciton. This is expected from the rapid intervalley exchange scattering of the photoexcited K-valley excitons into the K’ valley (Fig.2a-K’ valley).”

a. It is not clear if these excitons in K’ valley are bright or momentum-dark. Based on the inset of Fig. 2a, I believe the authors mean to communicate that the bright excitons in K scatter into bright excitons states in K’, but if this is the case, why is there no depletion of holes in K’?

b. Based on the data that I see in Fig. 2a, it appears that there are bright excitons at K, and momentum dark excitons in K’, with electrons at the K’ and holes in K. This would explain why we do not see the depletion of holes at K.

i. Based on the dynamics in Fig. 3b, it seems that there should be a significant amount of K-K’ indirect (I assume this is the momentum dark at K’? unclear) excitons even at 0.1 ps.

Reply to 2.a b and i

At early time delays, the signal in the K’ valley is indeed a contribution of both bright K’ excitons and K’-K momentum-dark exciton (electron in lower K’ state and hole in K). First, at 0.1 and 1 ps, we rule out any significant contribution from a momentum dark state with electron in the upper K’ state and hole in the K valley as it requires a process that flip only the exciton electron spin. Our analysis (see SI 7) shows a scattering time of 6 ps to this state, an order of magnitude slower than exchange interaction (0.35 ps) that forms K’ bright excitons and slower than intervalley scattering of the exciton-electron to the lower K’ state (0.8 ps) that forms K’-K momentum dark excitons. (We have provided additional new data and showing (see SI 7 and reply to Reviewer #2 point 3) that, at higher temperature the scattering of the exciton electron to the upper energy K’ state is strongly enhanced, which contributes to the overall loss of valley polarization shown in Fig.4.)

From the fit based on rate equations shown in Fig.3, we extract, at 0.1 ps, a bright K’ exciton density of $5 \times 10^{10} \text{ cm}^{-2}$ and a K’-K momentum-dark exciton density of $2 \times 10^{10} \text{ cm}^{-2}$, so the main

contribution in the K' valley originates from bright K' excitons. The larger contribution of bright excitons in the K' valley at photoexcitation can also be clearly seen in the ARPES data as shown in the revised Fig.2a (see reply to point 3a below), as most of the electron signal appears at the energy of the bright exciton.

Regarding the momentum distribution of holes in Fig.2a (revised Fig.2b, see in reply to point 3a below), our experimental signal-to-noise makes it challenging to visualize a well-defined momentum-resolved distribution for densities lower than 10^{11} cm⁻². To perform any quantitative analysis, we had to momentum and energy integrate the electron signals, allowing us to gain signal-to-noise and to obtain the data points for the holes of Fig.3a.

Change in manuscript:

- In the paragraph above Fig.2, we have added:

Additionally, at these early time-delays, we observe a weak signal in the K' valley at the energy of the bright exciton (see Fig.2a). This is expected from the rapid intervalley exchange interaction of the photoexcited K-valley excitons into the K' valley (Fig.2b-K' valley). We rule out any significant contribution from a momentum dark state with electrons in the upper K' state as it requires a spin-flip scattering process (enhanced at higher temperature, see SI § 7). The weakly appearing (kx, ky) momentum distribution of holes in Fig.2b in the K' valley is due to too low experimental signal-to-noise for this low density ($\sim 7 \times 10^{10}$ cm⁻²).

- In the first paragraph of the section “Dynamics of the long-lived valley polarized momentum dark exciton”, we have added:

To do so, we resolve the valley and spin states (via our energy resolution – Fig. 2a) of the exciton-bound electrons and holes over the entire BZ. The electron and hole populations are obtained by energy and momentum integrating their respective signals (see SI § 3 and § 6).

c. In Fig 2, the authors differentiate the intensity from bright and momentum-dark excitons by showing intensity at the excitonic energy and the depletion of intensity at the valence band. This analysis relies on the fact that a pump-induced change of the occupation function is the primary contributing factor to the photoemission intensity, and that the photoemission matrix element and spectral function have no explicit time dependence.

Typically, the spectral weight is conserved, such that an integration of the population and depletion gives a zero (see studies of TR-ARPES on graphene and graphite, for instance). If one integrates the electron population and hole depletion at both the K and K' point, is the spectral weight conserved? If the excitons at K' are meant to be

bright, then I would guess not? Why not? It would be nice to add consistent numbers to the colorbars here. d. If the spectral weight is not conserved, is the ratio of depletion/population the same at K and K', as one might expect if they are both bright excitons?

The reviewer is correct that one should carefully analyze ARPES signals, in particular when one wants to perform a comparative quantitative analysis between different valleys as well as different energies. In TR-ARPES, to accurately evaluate the density of photoexcited populations, we need to take into account the presence of momentum and energy varying photoemission matrix elements that can scale from 0 to 1. The photoemission matrix elements depend essentially on the energy, polarization and angle of incidence of the XUV probe. As shown in SI section 4 (reproduced below), the variation of intensity in the band structure is due to the variation of the photoemission matrix elements.

To ensure that we can compare the absolute densities between K and K' valleys, we perform two important steps:

- (1) As the reviewer highlighted, we carefully rotated our sample with respect to the XUV probe angle of incidence so the signals of the exciton-electrons and valence bands from two adjacent K and K' valleys are balanced.
- (2) The reviewer is correct that the spectral weight, although invariant with photoexcitation and time delay, is not the same at the energy of the holes and at the energy of the exciton-electrons due to the energy dependence of the photoemission matrix element. To access the absolute density, we use the hole depletion. The absolute hole density is obtained by subtracting the photoelectron counts between the unpumped valence band and the photoexcited one, after energy integrating over the VB linewidth and using a $0.1 \times 0.1 \text{ \AA}^{-1}$ area (as explained in SI section 6). This subtraction cancels the contribution of the matrix elements, providing the absolute 2D hole density. Then, we equalize (either using a linear excitation or at the rising edge of photoexcitation when all the photoexcited population is in one valley) the electron signals to the extracted hole density, thus eliminating the energy dependent contribution of the photoemission matrix elements.

All the steps described above allow us to have balanced signals between K and K' valleys, i.e. same electron counts correspond to an identical density, and to extract an absolute density allowing to circumvent the energy dependent variation of spectral weight as the electron counts are calibrated with respect to the absolute density extracted from the hole depletion. (Here we assume that the matrix element varies minimally over the exciton-electron energy distribution linewidth). We now provide a more detailed description of the procedure described above in the SI. Additionally, a detailed description of the hole density

extraction is below in “Dynamics of the long-lived valley polarized momentum dark exciton”, point 1.

Fig.R6: (Reproduced from Fig.S4) 2D ARPES data at the A exciton energy and at zero time delay (a) before sample rotation and (b) after a 10° rotation. (c) Photoemission intensity along a K-K' cut (orange dashed line in (a)) before rotation and (d) after rotation showing the equalization of intensity in the K and K' valleys under a linearly polarized photoexcitation. (e) Corresponding dynamics of the photoelectrons in the K and K' valleys showing intensity equalization at all time delays.

Change in SI section 6:

-We have added the following comment:

Taking the difference between unpumped and with pump data allows to eliminate the contribution from the photoemission matrix elements and to extract the absolute hole density. To obtain the electron densities, we assume that the photoemission matrix elements do not vary with time delay and we equalize the matrix elements between two adjacent valleys (See SI § 4). Then, we equalize the electron density to the previously extracted hole density using a linear polarization excitation, for which valley have a symmetric distribution (see SI § 4), and during photoexcitation. This allows us to account for the energy dependent variation of matrix element, assuming no variation within the electron signal energy linewidth.

3. Line 155: The authors write: “Strikingly, at 1 ps, we find that the dominant excitonic species is now a valley-polarized momentum forbidden exciton (K' - K exciton), as seen by the large electron population in the K' valley and a large hole population remaining in the K valley (Fig.2b).” While I do not disagree with this statement, the timescale of 0.1 ps and 1 ps is not so different. How do we know the ratio of bright excitons in K' with momentum-dark excitons at K' ?

We chose to display 0.1 ps and 1 ps because 0.1 ps time delay corresponds to the maximum of the photoexcited population (end of the pump pulse) in the K valley, corresponding to the maximum population reached by the K -valley bright excitons. At 1 ps, both the K and K' valley bright excitons already recombined as their lifetime is about 300 fs (as shown in Fig.3) so the remaining populations are exclusively composed of dark excitons (see also reply to next question). In Fig.2, at 1ps, we observe that most of the holes are located in the K valley and electrons in the K' valley, signature of the momentum-dark excitons. Qualitatively, the energy and momentum distribution of electrons as well as the momentum hole distribution allows us to understand what state dominates, and ruling out, at this time delay contributions from the scattering to excitonic states that require a spin-flip (see reply in point 2a, b, I and reply to next point). To obtain the exact ratio, we performed the analysis based on rate equations as in Fig.3 which confirms that, at this time delay, there is negligible contribution from bright states and that momentum dark states that accounts for 85% of the entire excitonic population.

a. Since Fig 2 shows data that is integrated from 1.9 eV to 2.2 eV, Fig. 1d is crucial to the interpretation of the data shown in Fig. 2, such that they should really be put together in one figure. Also, the panel of (0.1 ps K' valley) is missing from the story, and would be important to the interpretation.

We have now merged the energy resolved data previously in Fig.1d in Fig.2a. We thank the reviewer for this suggestion that improves the clarity of our manuscript. We also have added a panel (revised Fig.2a) for the energy and momentum resolved data for 0 ps at the K' valley which clearly shows that the main contribution comes from bright excitons. This new figure displays very clearly the respective contribution of bright and momentum dark excitons.

Fig.R7: Revised Fig.2. Valley polarized momentum dark excitons. (a) Energy and momentum resolved linecut along the Γ -K-M axis showing. At 0 ps time delay, the resonantly photoexcited bright exciton signal in the K valley shows an exciton-bound electron with negative dispersion, located 2.1 eV above the valence band (red). By 1 ps, in the same valley, this electron signal has relaxed to a lower energy state (blue). In the K' valley, a weak electron population is observed at 0 ps at the photoexcitation energy of 2.1 eV (green). At 1ps, it evolves into a much larger population that shows up at a lower energy (grey) (data around the exciton electron energy, 1.9 to 2.2 eV, were normalized at each k-vector). The corresponding energy distribution curves on the right clearly shows the energy difference between the bright exciton state which dominates at 0ps and lower energy states that shows up at 1ps. (b) Photoemission signals from electrons around the A exciton energy and from holes at the valence band during

photoexcitation (0.1 ps). For the electrons, the ARPES signals were energy integrated between 1.9 to 2.2 eV and displayed in k_x, k_y momentum space. For the holes, we display the difference between negative time delay and after photoexcitation ARPES signals at the top of the valence band. The data were energy integrated over 100 meV (-0.05 eV to 0.05 eV) and a 120° rotating average around the center of the valley was performed to clearly display the photoemission count loss corresponding to the presence of holes. (c) Photoemission signals from exciton bound electrons and holes around the A exciton energy at 1ps using a similar analysis as in (b).

Change in manuscript:

- Figure 2 has been replaced as shown above
- Figure 2 captions have been updated as shown above

4. The colors in Fig 2 are also confusing:

a. Previously red and blue were used for spin polarization, now they are used for depletion and population? Or valence band and conduction band?

b. Why do the insets have different colors? Yellow, red and blue? Are they meant to connect to something?

c. The insets are hard to see.

Reply for 4a-c:

We have now reformatted the figure to be consistent with the new color code and design used for the exciton diagram in Fig.1. We hope to have improved the readability of Fig.2.

d. Why is the integration region 300 meV for the excitonic states, but only 100 meV for the valence band?

In revised Fig.2b, to obtain the momentum resolved visuals, we have integrated the exciton-electron between 1.9 to 2.2 eV as it covers their entire energy distribution as seen in revised Fig.2a. For the valence band, we limited the energy range to a slice corresponding to the linewidth of the valence band. This allows to obtain a visual of the holes as seen in Fig.2. Integrating over a larger energy range adds the experimental noise in the vicinity of the band to the subtraction process visually smearing the hole contribution. We note that this energy range was used only to obtain the visuals of the holes in Fig.2. To obtain the absolute hole density, we used the same range as for the exciton-electron to capture the entire loss of counts.

5. Line 171: typo for “interaction”

Thank you for noticing. The typo has been corrected.

6. Line 171: The authors attribute the lack of valley-depolarization to intervalley exchange, which does not affect zero-momentum excitons. Can the authors give more support to this argument, in particular from experiment? Are there any other possible explanations for this?

We thank the reviewer for bringing this important point. At photoexcitation, the effect of increasing exciton density provides excess center-of-mass momentum and should result in an enhancement of the intervalley exchange interaction and thus to faster loss of valley polarization. To confirm this, we have performed additional experiments to measure the K-valley bright exciton valley polarization at 0.1 ps (corresponding to the maximum population of the K-valley bright exciton) for a range of density from $4.5 \times 10^{11} \text{ cm}^{-2}$ to $4 \times 10^{12} \text{ cm}^{-2}$. First, we have extracted the momentum distribution of the electron signals (See Fig.R8), and, as density increases, we observe a broadening of the momentum distribution that is due to an increase of center-of-mass momentum as previously shown in [Sci. Adv. 7, abg0192 (2021)]. Then, we extracted the corresponding valley polarization of the K-valley bright exciton at the same time-delay, and we clearly see a decrease of valley polarization associated to the increase of center-of-mass momentum, as one would expect from enhanced intervalley exchange interaction.

Fig.R8: (a) Momentum distribution of the exciton-electron signals at low and high density. (b) Momentum width of the exciton-electron signal (black squares) compared to the K-valley bright exciton valley polarization (red line).

Change in manuscript:

- In the paragraph above Fig.4, we have added:

A higher intensity (Fig.4c and 4d) provides excess center-of-mass momentum to the bright exciton that is expected to enhance of the intervalley exchange interaction³⁴ and lead to a faster depolarization. This is confirmed by measurements showing that increasing pump intensity leads to a reduced valley polarization of the bright exciton, accompanied by a broadening of the exciton–electron momentum distribution (See SI § 10).

Change in SI:

- The discussion above and corresponding figure has been added to SI section 10

Dynamics of the long-lived valley polarized momentum dark exciton

1. The supplementary material mentions that the electron and hole populations are partitioned the K-up, K-down, K'-up and K'down states. But does not mention which energy-momentum windows from the experimental data are integrated for these values, and this is quite crucial to the analysis. Could the authors provide some spectra and show which intensities are calculated from which regions?

We hope that our reply to point 2.c has already clarified our method to extract exciton densities. We used a 0.1×0.1 Å⁻¹ area centered at the K and K' valleys. To obtain the density we have integrated over the same energy width the electron signals and valence band signals, i.e. 400 meV (we note that we use here a different energy range for the valence band than the 100 meV used to visualize the hole distribution of Fig.2b). In the figure below, we illustrate this for zero-time delay by displaying in the bandstructure the momentum-energy range used for our analysis. We also provide the associated spectra for the electrons in the K and K' valleys. In the panel c and d, we show the spectra (here normalized with respect to the lower energy valence band) comparing unpumped and pumped data showing at zero-time delay the depletion of counts in the upper valence in the K valley and very minimal change in the K' valley. The corresponding hole depletion corresponds to a total loss of 2695 counts or a fraction of 6% in the K valley (414 counts or 0.9% fraction in the K' valley) of the total density (by taking the ratio with unpumped data). From here, the density can be calculated by multiplying the fraction with the constant 2D density of states for the area used resulting in a density of 4.5×10^{11} cm⁻² in the K valley and 7×10^{10} cm⁻² in the K' valley.

Fig. R9: (a) Experimental bandstructure showing the energy-momentum window used for the data analysis (white rectangles). The signal above 0.4 eV have been multiplied by 100 for clarity. (b) Electron signal at 0 ps showing the contrast between K and K' valley bright excitons. (c, d) Energy distribution curves of the valence band showing majority depletion of holes in the K valley compared to the K' valley.

Change in SI:

- Fig.R9 has been added as Fig.S7 of the SI section 6
- The above discussion has been added to the same section

2. Line 215: How is degree of circular polarization (DOCP) calculated? And is this related to valley polarization?

In optical spectroscopy, the degree of circular polarization is commonly used to quantify the valley polarization and is defined as the intensity contrast between the emission that is right- circularly ($\sigma+$) and left- circularly ($\sigma-$) polarized such that the $DOCP = (I_{\sigma+} - I_{\sigma-}) / (I_{\sigma+} + I_{\sigma-})$. This definition originates from the interband optical selection with circularly polarized light. In our work, the degree of valley polarization for the momentum dark exciton is defined as

$P(X_{K'-K}) = \frac{nX_{K'-K} - nX_{K-K'}}{nX_{K'-K} + nX_{K-K'}}$ where nX_i is the density of exciton X_i . For clarity we are replacing this term by “degree of valley polarization” and explicitly define it in the manuscript. .

Change in manuscript:

- We have replaced the term DOCP by “degree of valley polarization” in the paragraph above Fig.3 and in the conclusion.
- We have added the definition:

This $K'-K$ momentum dark exciton remains the dominant species across our experimental temporal range (10 ps) and maintains a high degree of **circular valley** polarization (>40%), defined as $P(X_{K'-K}) = \frac{nX_{K'-K} - nX_{K-K'}}{nX_{K'-K} + nX_{K-K'}}$, where nX_i is the density of exciton X_i , (~~DOCP~~) through this time (Fig. 3c).

3. Figure 3 is also not entirely clear. In particular the insets are again small and the text is difficult to read. The diagram to the right of Fig. 3b is also not entirely clear with respect to the dual arrows. For example, is exchange + phonon interaction directly converting the bright exciton to the K' spin-dark exciton? Or is there an intermediate state in the K' bright exciton? Or is it both? a. To alleviate some heavy unreadable text, the authors may consider giving a symbol to each exciton, for instance “ K_b ” for “bright exciton at K ” and “ $K's$ ” for “spin-dark exciton at K' ”.

We have redesigned Fig.3 for readability according to the reviewer’s suggestions. The excitons are now labeled throughout the manuscript and figures as X_{K_b} and $X_{K'_b}$ for bright excitons, X_{K-K} and $X_{K-K'}$ for the momentum dark excitons and X_{K_s} and $X_{K'_s}$ for the spin-dark excitons. The arrows have been redesigned to avoid confusion in the sequence of formation of the excitons.

Fig.R10: Revised Fig.3. Valley-polarized bright exciton to dark exciton scattering dynamics and distribution at low temperature and low exciton density. (a) Measured dynamics of the electron density in the K and K' valley for each spin-split state and hole density in the valence band (dots). The dotted lines show the fit obtained from our model based on rate equations. (b) Bright, momentum-forbidden and spin-forbidden dominant exciton populations extracted from our model based on the experimental fit in (a). At 1ps, we display each excitonic species contribution to the total population. On the right, we present a diagram describing the formation and scattering processes associated to each excitonic populations. Inset: Normalized early time dynamics showing the formation sequence of each excitonic state. (c) Temporal evolution of the degree of circular polarization and relative population ratio for the bright excitons (yellow) and for the $X_{K'-K}$ and $X_{K-K'}$ momentum-dark excitons (blue).

4. Line 240: Given that photoexcitation intensity is crucial, do the authors have any fluence dependence that can show how the valley polarization evolves with the density of excited carriers? Such a dependence may give clues as to the interactions involved in valley-depolarization.

We have addressed this point in point 6 of the section “TR-ARPES experiment on 1L WS2: Valley-selective excitation” above.

REVIEWER RESPONSE

Reviewer #1:

The authors have well responded to my comments. The novelty of this manuscript is now clearly demonstrated, and I would like to recommend its publication in Nature Communications.

We are delighted to have addressed all the reviewer's comments and that the novelty shown in our revised manuscript should be published in Nature Communications.

Reviewer #2:

The authors have now provided a more explanation of the novelty of their work and clarified the specific knowledge gaps their study addresses, particularly in reconciling discrepancies between prior optical and TR-ARPES observations of dark exciton valley polarization. I appreciate these clarifications and the improvements made to the manuscript.

While the technical execution and analysis in this manuscript are strong enough, the scientific significance is not transformative on its own, relative to the expectations of Nature Communications. The central result — resolving time-dependent valley polarization dynamics of different dark exciton species via TR-ARPES — is certainly valuable, but remains incremental, building closely on prior work, including the authors' own.

For publication in a general-interest journal with a broad scientific scope, such as Nature Communications, a demonstration of dark exciton dynamics alone may not be sufficient. To meet the journal's criteria, I would expect the authors to go further and demonstrate some form of active control, tunability, or a compelling connection to practical device-level implementation using such long lived dark exciton, which would significantly enhance the manuscript's impact.

As it stands, the manuscript reads as a technically well-executed diagnostic study, which may be more appropriate for a specialized journal focused on ultrafast

spectroscopy.

I encourage the authors to consider extending the scope of their work to demonstrate how these findings could be leveraged for real-world applications, in order to fully realize the potential impact and completeness of the study.

We respectfully disagree with the reviewer. To begin with, the reconciliation between two powerful experimental platforms – optics and ARPES – in their observations of important properties of 2D materials, will be of vital interest to readers in ARPES, photoemission spectroscopy, optics, and the broad 2D material community. Given the importance of these two experimental platforms in understanding materials science and condensed matter physics (CMP) in general, a concrete and important example of how the two techniques may disagree, and how they could be reconciled, will be of value to the broader material science and CMP communities themselves, beyond 2D materials.

Next, besides the reconciliation, the knowledge gaps that we have filled, regarding the nature and timescales on how different valley polarized dark excitons contribute to the overall valley polarization, are of importance to the 2D valleytronics community and those seeking to develop quantum technologies and devices using 2D materials, particularly with decoherence protected dark excitons. The information we report will help these communities devise specific schemes to manipulate the relevant dark excitons and the information they carry.

Additionally, as the reviewer has pointed out, our results in time-resolved imaging of the entire Brillouin Zone and measuring exciton dynamics are also of interest to the broader ultrafast community.

Overall, given the breadth and variety of communities that will benefit from our study, we believe a top, interdisciplinary journal like Nature Communications is indeed the ideal venue to disseminate our results.

Finally, in the context of the Reviewer's request to demonstrate control, we would like to point out that our results indeed demonstrate control over the valley-polarization of dark excitons - by controlling the experimental parameters of intensity, pump wavelength and temperature, we show that one can switch between a fast valley-depolarization process and one where valley-polarized dark excitons hold information for long times.

To clarify this aspect of our work, we added a comment in the discussion section.

Change in manuscript:

In the Discussion section, we have added the following:

Our work shows that, depending on experimental conditions, one can switch from a rapid depolarization process to the formation of long-lived valley-polarized dark excitons.

Reviewer #3:

The authors have replied satisfactorily to my concerns and I support the publication of this manuscript. Below I leave some comments for the authors to consider.

General Comments

- k_x and k_y should have subscripts.

Detailed Comments

- Fig 1c caption “Exciton-bound electron energy” sounds strange and doesn’t really make sense. Perhaps the authors should quote the energy above the valence band and expand on what that energy region represents in the text.
- Line 142: “momentum dark” is missing a hyphen.
- This discussion of Fig. 2 is hard to follow because there are 9 panels in Fig. 2a and 8 panels in Fig. 2b. Could the authors provide some additional labels and/or color references to track which panels go together between a and b and refer to them in the text? For example (line 149): we observe a weak signal in the K' valley at the energy of the bright exciton (see Fig.2a, green dashed line)
- Fig 3 caption: XKs is not defined

We thank the Reviewer for their support and for the final suggestions and corrections.

Change in manuscript:

- All typos described above have been corrected.
- Fig.1c caption now clarifies the energy integration of the electron signal.
- All panels of Fig.2 have been relabeled for clarity. Changes have been made in the text accordingly.
- XKs are now defined in the caption of Fig.3.

We conclude by thanking all three reviewers once again for their very useful feedback on our work.